

# Early Eocene vigorous ocean overturning and its contribution to a warm Southern Ocean

Yurui Zhang[1], Thierry Huck[1], Camille Lique[1], Yannick Donnadieu[2,3], Jean-Baptiste Ladant[4], Marina Rabineau[5], Daniel Aslanian[6]

[1]Univ Brest, CNRS, IRD, Ifremer, Laboratoire d'Océanographie Physique et Spatiale (LOPS), IUEM, Brest, France

[2] Laboratoire des Sciences du Climat et de l'Environnement, LSCE-IPSL, CEA/CNRS/UVSQ, Université Paris-Saclay, Gif-sur-Yvette, France,

[3]Aix Marseille Univ, CNRS, IRD, INRA, Coll France, CEREGE, Aix-en-Provence, France,

[4] Department of Earth and Environmental Sciences, University of Michigan, Ann Arbor, MI, USA

[5] CNRS, Laboratoire Géosciences Océan (LGO, UMR 6538 CNRS, Univ Brest, Univ Bretagne-Sud), IUEM, Plouzané, France

[6] IFREMER, Unité de Recherche Géosciences Marines, Centre de Bretagne, Plouzané, France

*Corresponding author: Yurui Zhang (yurui.zhang@univ-brest.fr)*

**Abstract.** The early Eocene (~55 Ma) is the warmest period, and most likely characterized by the highest atmospheric
$CO_2$ concentrations, of the Cenozoic era. Here, we analyze simulations of the early Eocene performed with the IPSL-
CM5A2 coupled climate model set up with paleogeographic reconstructions of this period from the DeepMIP project,
with different levels of atmospheric $CO_2$, and compare them with simulations of the modern conditions. This allows
us to explore the changes of the ocean circulation and the resulting ocean meridional heat transport. At a $CO_2$ level of
840 ppm, the Early Eocene simulation is characterized by a strong abyssal overturning circulation in the Southern
Hemisphere (40 Sv at 60ºS), fed by deep water formation in the three sectors of the Southern Ocean. Deep convection
in the Southern Ocean is favored by the closed Drake and Tasmanian passages, which provide western boundaries for
the build-up of strong subpolar gyres in the Weddell and Ross seas, in the middle of which convection develops. The
strong overturning circulation, associated with the subpolar gyres, sustains the poleward advection of saline
subtropical water to the convective region in the Southern Ocean, maintaining deep-water formation. This salt-
advection feedback mechanism works similarly in the present-day North Atlantic overturning circulation. The strong
abyssal overturning circulation in the 55 Ma simulations primarily results in an enhanced poleward ocean heat
transport by 0.3–0.7 PW in the Southern Hemisphere compared to modern conditions, reaching 1.7 PW southward at
20°S, and contributing to maintain the Southern Ocean and Antarctica warm in the Eocene. Simulations with different
atmospheric $CO_2$ levels show that the ocean circulation and heat transport are relatively insensitive to $CO_2$-doubling.
**Keywords:** Early Eocene, overturning circulation, deep water formation, oceanic heat transport.





## 1. Introduction

Proxy-based temperature reconstructions suggest that the early Eocene (55–50 Ma) was one of the warmest intervals in the geological history and the warmest of the Cenozoic (Zachos et al., 2001; Cramer et al., 2011; Dunkley Jones et al., 2013). More specifically, the EECO (Early Eocene Climatic Optimum) occurred 51–53 Ma, but shorter (less than tens of thousands of years) hyperthermal events such as the PETM (Paleocene-Eocene Thermal Maximum) about 55 Ma ago (Zachos et al., 2008) also occured. The Southern Hemisphere was particularly warm at that time, as shown by inferred surface ocean temperatures exceeding 20°C at high-latitudes (e.g. Evans et al., 2018 and references therein), and by the absence of a perennial ice over Antarctica until the onset of the Antarctic ice sheet at the Eocene-Oligocene Boundary, ~34 Ma when $CO_2$ abruptly declined below a certain threshold (Galeotti et al., 2016; Gasson et al., 2014; Ladant et al., 2014). In the early Eocene, high levels of $CO_2$ in the atmosphere are undoubtedly a critical contributor to the extremely warm climate, with the global temperature increasing by more than 5°C in less than 10 000 years (Zachos et al., 2001, 2008; Huber and Caballero, 2011; Anagnostou et al., 2016), but they do not fully explain the extreme warmth at high-latitudes and the reduced equator-to-pole temperature gradient (Huber and Caballero, 2011).

In addition to the much higher levels of $CO_2$ in the atmosphere, one of the main differences between the early Eocene and our modern climate lies in the very contrasted bathymetries and continental configuration, likely resulting in very different ocean circulation (Thomas et al., 2003; Voigt et al., 2013; Winguth et al., 2012; Zachos et al., 2001). In particular, the opening or closing of major oceanic gateways (such as the Drake Passage or the Panama Seaway) during the Late Paleogene and Neogene have been shown to exert a strong influence on the ocean circulation and its associated heat transport (England et al., 2017; Ladant et al., 2018; Nong et al., 2000; Sijp and England, 2004; Toggweiler and Bjornsson, 2000; Yang et al., 2014). Additionally, proxy-based reconstructions and results from Eocene model simulations suggest that the Meridional Overturning Circulation (MOC) was also very distinct from our present day MOC, with no evidence for deep water formation in the North Atlantic until the early Oligocene (Ferreira et al., 2018). Instead, formation of deep water was found to happen in the North Pacific (Hutchinson et al., 2018; Winguth et al., 2012) or only in the Southern Ocean (Sijp et al., 2014). Different ocean circulation resulting from different bathymetry are expected to result in different Ocean Heat Transport (OHT). For instance, Sijp and England (2004) found a 0.5 PW decrease in OHT in the Southern Hemisphere in response to the opening of the Drake Passage, but other factors such as different radiative forcing induced by different level of $CO_2$ in the atmosphere may also contribute to the different OHT (see Huber, 2012 for a review).

In our present climate, the ocean is an important actor for the Earth energy balance, as it contributes about one third to the total redistribution of heat from the Equator to the Poles (e.g. Trenberth and Caron, 2001). Although modifications of both the atmosphere and the Ocean Heat Transport (OHT) tend to compensate (Trenberth and Caron, 2001), subtle changes in OHT could trigger large changes in atmospheric extratropical convection, modifying the water vapor greenhouse (Rose and Ferreira, 2013), which in turn can affect surface temperature. The OHT itself results from different contributions, and several attempts have been made in the literature to disentangle the relative roles of the horizontal and overturning ocean circulations for the meridional OHT (Ganachaud and Wunsch, 2003). In the North Atlantic, where the strong Atlantic Meridional Overturning Circulation (AMOC) is fed by the formation of





dense water by convection at high latitudes, the AMOC contributes up to 90% of the meridional OHT at 26.5ºN
(Msadek et al., 2013), where the RAPID monitoring array is located (McCarthy et al., 2015). Based on hosing
experiments performed with a climate model, Yang et al. (2013) found that the meridional OHT decreases rapidly in
response to an artificial shutdown of the AMOC, although Drijfhout and Hazeleger (2006) suggested that on longer
(decadal) timescale, the OHT might recover its initial level as the gyre contribution tends to compensate the decrease
in OHT associated with the AMOC shutdown. In contrast to the North Atlantic, Volkov et al. (2010) found that, in
the Southern Ocean, the OHT results roughly equally from the gyre and overturning contributions.
Given the importance of both horizontal and overturning ocean circulations for the meridional OHT, one can expect
that the different MOC and horizontal gyre constrained by the Eocene bathymetry would result in different
contributions to the meridional OHT, potentially contributing to the warm climate at 55 Ma, and in particular to the
much warmer temperatures found at high-latitudes. Based on the analysis of early Eocene and modern simulations
performed with the IPSL climate model, the goal of this study is to better understand what sets the ocean circulation
during the Eocene and the importance of the ocean circulation for the poleward heat transport. The model and
simulations used are described in Section 2. In Sections 3 and 4, we examine the MOC and the horizontal circulation,
respectively, for the present-day and 55 Ma configurations, and these circulations are then linked to the OHT in Section
5. The sensitivity of the ocean circulation and heat transport to the level of $CO_2$ in the atmosphere is discussed in
section 6. A summary and conclusions are given in Section 7.
**2. Numerical model data**
**2.1. Model setup and simulations**
The simulations used in this study are performed with the IPSL-CM5A2 earth system model (Sepulchre et al., 2019).
The oceanic component of IPSL-CM5A2 is NEMOv3.6 (Madec and the NEMO team, 2016), which includes the LIM2
sea ice model (Fichefet and Maqueda, 1997) and the PISCES biogeochemical model (Aumont et al., 2015). The
atmospheric component is the LMDz model (Hourdin et al., 2013), which is coupled to the land surface model
ORCHIDEE (Krinner et al., 2005). Here, we use IPSL-CM5A2 in its standard resolution. NEMO is thus run at a
nominal resolution of 2°, increased down to 0.5° at the equator, with 31 levels that vary in thickness with depth. LMDZ
is run at a horizontal resolution of 3.75° longitude×1.875° latitude and 39 vertical levels. A full description and
evaluation of the IPSL-CM5A2 model can be found in Sepulchre et al. (2019).
Two sets of simulations are performed. The first set is composed of the reference PI simulation (referred to as PI-1x
and described in Sepulchre et al., 2019) and another PI simulation in which the atmospheric $CO_2$ concentration is
doubled (PI-2x). The second set consists in a baseline simulation of the Early Eocene (hereafter 55 Ma-3x), using a
setup following the DeepMIP protocol described by Lunt et al. (2017), and two sensitivity experiments to $CO_2$ (55
Ma-1.5x) and tidal mixing (55 Ma-noM2). In the following, we first briefly describe the baseline Early Eocene
simulation (as the DeepMIP guidelines from Lunt et al. (2017) give several options for implementing the Early Eocene
boundary conditions), and then precise the boundary condition differences of the two other simulations.



### 2.1.1. General considerations

Most of the boundary conditions that were adapted for these IPSL-CM5A2 Early Eocene simulations are described in Herold et al. (2014, hereafter H14). Following Lunt et al. (2017), the solar constant and orbital parameters are kept to PI values, so are greenhouse gas concentrations with the exception of atmospheric $CO_2$. The latter is set to 840 ppm (3x the PI value), so that the simulation is representative of the pre-PETM (following the terminology of Lunt et al., 2017).

### 2.1.2. Oceanic boundary conditions

The ocean component (NEMO) is commonly run on a tripolar mesh grid (Madec and the NEMO team, 2016), which avoids singularity points in the ocean domain. Because the implementation of the Eocene land-sea mask on the ORCA2 grid would have shifted the singularity points into the ocean domain, we have constructed a new PALEORCA grid, which is suitable to run paleo-simulations with IPSL-CM5A2 (see more details in Sepulchre et al., 2019). The bathymetry is obtained by masking out the H14 topography and remapping the resulting bathymetry onto the PALEORCA grid using near-neighbor interpolation. This type of interpolation indeed ensures that small but crucial features of the H14 dataset such as islands and seaways, which may strongly impact the modeled ocean circulation, remain present in the interpolated bathymetry file. Handmade corrections were then applied at some locations (e.g. in the West African region) to retain oceanic straits that are sufficiently large to allow for exchanges. The Early Eocene bathymetry is shown on Fig. 1.

Modern boundary conditions of NEMO also include forcings of the dissipation associated with internal wave energy from the M2 and K1 tidal components (de Lavergne et al., 2019). The parameterization follows Simmons et al. (2004), with refinements in the modern Indonesian Through Flow (ITF) region according to Koch-Larrouy et al. (2007). To create an Early Eocene tidal dissipation forcing, we directly interpolate the H14 M2 tidal field (obtained from the tidal model simulations of Green and Huber 2013) onto the NEMO grid using bilinear interpolation. In the absence of any estimation of the K1 tidal component for the Early Eocene, we ignore this contribution. In addition, the parameterization of Koch-Larrouy et al. (2007) is not used here because the ITF does not exist in the Early Eocene.

The geothermal heating distribution $q$ is created from the 55 Ma global crustal age distribution of Müller et al. (2008), on which the age-heatflow relationship of the Stein and Stein (1992) model is applied:

$$q(t) = 510\, t^{-1/2} \qquad t \leq 55 \text{ Ma}$$

$$q(t) = 48 + 96\, e^{-0.0278\, t} \qquad t > 55 \text{ Ma}$$

In regions of subducted seafloor where age information is not available, we prescribe the minimal heatflow value derived from known crustal age. The 1°x1° resulting field is then bilinearly interpolated on the NEMO grid. It must be noted that the Stein and Stein parameterization becomes singular for young crustal ages, which yields unrealistically large heatflow values. We thus set an upper limit of 400 mW.m$^{-2}$ on heatflow values, following Emile-Geay and Madec (2009).

Salinity is initialized as globally constant to a value of 34.7 psu following Lunt et al. (2017). The initialization of the model with the proposed DeepMIP temperature distribution (Lunt et al., 2017) led to severe instabilities of the model during the spin-up phase. The initial temperature distribution has thus been modified to follow:





$T\,(°C) = \frac{(1000-z)}{1000}\,25\,cos\,(\varphi) + 10$     z ≤ 1000 m
$T\,(°C) = 10$                         z > 1000 m
with $\varphi$ the latitude and z the depth of the ocean. This new equation gives an initial globally constant temperature of
10°C below 1000 m and a zonally symmetric distribution above, reaching surface values of 35°C at the equator and
10°C at the poles. This corresponds to a 5°C surface temperature reduction compared to the DeepMIP equation (Lunt
et al., 2017). No sea ice is prescribed at the beginning of the simulations.
The IPSL-CM5A2 model includes PISCES biogeochemical model. Biogeochemical cycles and marine biology are
directly forced by dynamical variables of the physical ocean and may affect the ocean physics via their influence on
chlorophyll production, which modulates light penetration in the ocean. However, because this feedback does not
affect the ocean state significantly (Kageyama et al., 2013) and because the Early Eocene mean ocean color is
unknown, we have prescribed a constant chlorophyll value to 0.05 g.Chl/L for the computation of light penetration in
the ocean. As a consequence, marine biogeochemical cycles and biology do not alter the dynamics of the ocean and
as such, biogeochemical initial forcings have been kept to modern.
2.1.3. Continental boundary conditions
The atmospheric (LMDZ) and land surface (ORCHIDEE) models run on a low resolution grid but require input
forcings at higher resolution. The topographic field is created by masking out ocean points in the H14 reference file
and upscaling the 1°x1° masked H14 file to the required LMDZ input topographic resolution (1/6°), as LMDZ includes
a subgrid scale orographic drag parameterization requiring high-resolution surface orography (Lott and Miller, 1997;
Lott, 1999). A similar procedure is applied to the standard deviation of orography proposed by H14.
Following Lunt et al. (2017), the soil properties are prescribed as globally constant to the global mean of the PI
simulation. There is no lake module in this version of IPSL-CM5A2. The river routing proposed by H14 is passed to
ORCHIDEE at its original resolution of 1°x1°, which ensures an appropriate downscaling to the model resolution.
The vegetation cover is prescribed from the BIOME4 reconstruction of H14, using a lookup table (given in Table S2)
to convert the 10 megabiomes into ORCHIDEE Plant Functional Types (PFTs). Aerosol distributions are left identical
to PI values.
2.1.4. Sensitivity experiments and equilibrium
We perform two additional Early Eocene experiments. One has the same boundary conditions as the baseline Early
Eocene experiment (55 Ma-3x) but an atmospheric $CO_2$ concentration of 420 ppm (1.5x the PI levels, 55 Ma-1.5x).
The other differs from the baseline Early Eocene experiment by the absence of tidal dissipation forcing (55 Ma-3x-
noM2).
The 55 Ma-3x simulation is initialized from rest and run for 4000 years (Fig. S1). The 55 Ma-1.5x simulation is
branched from 55 Ma-3x at year 1500 and run for 4000 years. The 55 Ma-3x-noM2 is branched from 55 Ma-3x at
year 3000 and run for 2000 additional years. The two PI simulations are initialized from the Levitus climatology
(Boyer et al., 2005), and run for more than 2700 years (Table 1). At the end of all the simulations, the ocean has



reached a quasi-equilibrated state and trends in deep ocean temperatures over the final 1000 years of all simulations
are smaller than 0.05°C/century.
We use the monthly outputs of the last 100 years of each simulation to create a climatological year for each simulation.
In the following, we will mostly focus on the comparison between the baseline Early Eocene simulation (55 Ma-3x)
and the PI control simulation (PI-1x). The other simulations are analyzed in Section 3 to estimate the contribution of
tidal mixing to the oceanic overturning circulation, and in Section 6 to examine the sensitivity of the ocean conditions
to different levels of $CO_2$ in the atmosphere.

**2.2. Evaluation of the simulated ocean temperature**

The mean state and the seasonal variation of ocean temperature in the 55 Ma simulations are examined. Further, we
evaluate the ability of IPSL-CM5A2 to reasonably simulate the early Eocene sea surface temperature (SST).
The annual mean SST in the 55 Ma-3x simulation varies from 10–15°C in the Southern Ocean, to 37.2°C near the
Equator (Fig. 2A), with a global mean of 27.5°C. During summer (defined as July-August-September for the Northern
Hemisphere and January-February-March for the Southern Hemisphere), the simulated SST reaches ~20°C over most
of Southern Ocean (south of 60°S), and up to 38°C in parts of equatorial Indian and Atlantic oceans (Fig. 2A). In the
55 Ma-1.5x simulation, both SST and global mean temperature are ~5°C lower than in the 55 Ma-3x simulation (Table
173 1).

These simulated SSTs are further compared with proxy-based SST estimates for the early Eocene provided by a recent
data compilation performed within the DeepMIP framework (Hollis et al., 2019). The dataset includes 32 records in
total, from 4 proxy types (TEX[86], $\delta^{18}O$, Mg/Ca and Clumped Isotope data). The spatial pattern of the model SST is
overall consistent with the proxy based SST, although significant differences can be seen for some specific proxy data
point (Fig. 2A and S2A). More details on the model-proxy comparison can be found in Supplementary Material.
In order to further compare the simulations with the proxy-based reconstructions, we also calculate the root-mean-
square deviations (RMSD) between the simulated SST and the reconstructions (Table 2). Although large, the RMSD
values are overall of the same order of magnitude as the uncertainty of proxy-based SST estimates, suggesting a
reasonable model-data consistency. More importantly, the RMSD values are smaller for the 55 Ma-3x simulation than
for the 55 Ma-1.5x simulation, suggesting that the 55 Ma-3x simulation captures better the signal of proxy-based SST
reconstructions. This is also consistent with proxy reconstructions suggesting that the $CO_2$ atmospheric concentrations
during the early Eocene were most likely three to four times the PI level (Foster et al., 2017).
The zonal mean SSTs in the 55 Ma-3x simulation range from 30°C to 37°C in the tropics and decreases toward the
high latitudes (Fig. 2B). Within the 40°S–40°N latitudinal band, the summer SSTs remain above 30°C. Around 60–
70°S the annual mean SSTs are ~13°C with a seasonal amplitude of 10 to 15°C. Those zonal mean SSTs are overall
~10°C warmer than in PI-1x simulation, with the largest differences of 12°C found in the Southern Ocean (Fig. 2B).
The warm SSTs found in the Southern Ocean in the 55 Ma simulations also extend at depth (Fig. 2C), with a mean
global temperature of 11.3°C in the 55 Ma-3x run, compared to 3.3°C in PI-1x (Table 1). The very warm temperatures
found at depth are compatible with several proxy-based temperature estimates (bottom temperatures at 1000-5000m



between 10 and 15°C, e.g. Huber et al., 2000, their Fig. 4-1). More specifically, Mg/Ca-based temperature estimates
suggest that the bottom-water (below 1000 m) temperatures were around 15°C during the early Eocene (Cramer et al.
2011), and Dunkley Jones et al. (2013) found a similar value based on $\delta^{18}$O-Mg/Ca thermometry data for the PETM
time-window.

**3. The overturning circulation**

Here we describe the simulated MOC in the different simulations, and investigate the links between the MOC and
deep-water formation.

**3.1. Meridional overturning circulation**

The global MOC is represented through the vertical streamfunction ψ computed from the zonally-integrated
meridional volume transport as:

$$\psi(z) = \int_W^E \int_{-H}^0 v(x,z) \, dz \, dx \qquad (1)$$

where v is the meridional velocity, z is the vertical coordinate, H is the ocean depth, x is the zonal coordinate
integrated from West (W) to East (E) boundary.
In the 55 Ma-3x simulation, a single clockwise inter-hemispheric MOC cell fills the whole deep ocean, with a
maximum of 40 Sv at 1500 m depth and 60°S (Fig. 3). This strong MOC cell (referred to as SOMOC for Southern
Ocean MOC) is associated with the formation of (Eocene) Antarctic Bottom Water (AABW) in the Southern Ocean,
flowing northward below about 2000 m. The SOMOC is associated with an upwelling branch extending over the
whole Northern Hemisphere, with almost 26 Sv crossing the equator northward (22 Sv at 20°N and only ~5 Sv at
60°N). There is no deep-water formation in the Northern Hemisphere, neither in the North Atlantic nor in the North
Pacific.
In contrast to the 55 Ma-3x run, the MOC in the PI-1x simulation is composed of the traditional upper and lower cells.
The upper cell is clockwise and associated with the AMOC, with a maximum strength of 11–12 Sv reached at 800 m
depth around 50–60°N. This cell is fed by the formation of North Atlantic Deep Water (NADW) around 60°N. NADW
is transported southward all the way to the Southern Ocean at depth between 1000 and 3000 m, where it is brought
back to the surface through wind-induced upwelling, forming the quasi-adiabatic pole-to-pole overturning circulation
regime (Marshall and Speer, 2012; Wolfe and Cessi, 2014). The lower cell is anticlockwise, with a maximum strength
of 15–16 Sv at 3000 m depth. This anticlockwise circulation is fed by the Antarctic Bottom Water formed by surface
buoyancy loss related to ocean-sea-ice interaction (Abernathey et al., 2016), and consumed through mixing process
induced by breaking of internal wave at the sea floor and geothermal heating (Nikurashin and Ferrari, 2013; de
Lavergne et al., 2016).
The 55 Ma simulated deep-water formation in the Southern Ocean is compatible with currently available proxy-based
reconstructions of the MOC. Although proxy data constraining the ocean circulation are very limited for the early





Eocene, these data seem to point to a common deep-water source around the Austral Ocean during that period (Abbott
et al., 2016; Batenburg et al., 2018; Frank, 2002; Thomas et al., 2003, 2014). For instance, Batenburg et al. (2018)
revealed a convergence of Nd-isotopes signature across the Atlantic basin from 59 Ma onward, likely resulting from
an intensification of the intermediate and deep ocean circulation in the Atlantic Ocean, with the dominant deep-water
masses originating from the high southern latitudes. In addition, the reconstructed deep-sea carbonate ion
concentration ($[CO_3^{2-}]$) during the Eocene shows a reversed inter-basin gradient compared to the present-day,
suggesting a reverse ocean circulation at depth compared to the present-day circulation (Zeebe and Zachos, 2007).
Therefore, these proxy data and the simulated ocean circulation are compatible, at least on the direction of the deep
circulation and the source of the deep water masses.
**3.2. Convection and deep-water formation**
The abyssal circulation described in Section 3.1 is fed by deep convection processes that mainly occur in winter. We
examine the simulated Mixed Layer Depth (MLD) at the end of the winter season (Fig. 4), which is an efficient
indicator of convection. In the 55 Ma-3x simulation, deep convection occurs only at high-latitude in the Southern
Hemisphere, with maximum MLD reaching up to 4000 m in the Weddell Sea, and large areas of MLDs deeper than
2000 m around Antarctica in the Ross and Amundsen seas (Fig. 4). In contrast, MLDs remain shallow in the Northern
Hemisphere, suggesting the absence of any deep convection there. In this hemisphere, the maximum MLDs are found
in the North Pacific (350 m) over the poleward western boundary current between 35–50°N, and in the North Atlantic
between 35–40°N (300 m), but the deepest MLD at high latitude are only 200 m over the northwest Pacific. Note that
the early Eocene North Atlantic basin is limited to a narrow region west of Greenland and poleward of 52°N. In the
absence of other sources of deep waters, waters sourced in the Southern Ocean around Antarctica fill the whole abyssal
ocean (Fig. 3).
In contrast, deep-water formation occurs both in the northern North Atlantic and in the Weddell Sea in the PI-1x
simulation (Fig. 4, right column). The deepest MLD (up to 1500 m) are found in the Nordic Seas between 70–75°N
and just south of Iceland around 60°N. MLD larger than 1200 m are also found in the eastern Weddell Sea around
65°S, indicative of the formation of AABW. This pattern is consistent with the MLD simulated by the CMIP5
ensemble for modern conditions (Heuzé et al., 2015).
In 55 Ma-3x, the permanent absence of deep convection in the North Pacific is at odds with a few previous model
studies and proxy-based reconstruction of the ocean circulation of the early Eocene that have suggested that deep
water could form in the North Pacific (Lunt et al., 2010; Abbott et al., 2016). The reasons why deep water forms in
the Southern Ocean in the 55 Ma-3x simulation and not in the North Pacific can be linked to the different ocean
thermo-dynamical properties of these two regions. Table 3 summarizes key ocean surface variables in the convective
regions of the Weddell Sea, where the deepest MLD are found, and of the North Pacific. These two regions roughly
correspond to closed Sea Surface Height (SSH) contours around deep convection regions, which are associated with
a minimum of SSH— the geostrophic ocean surface currents circulate cyclonically along the SSH contours, so that
the interior regions remain largely isolated from surrounding waters. The Weddell Sea is characterized by a strongly



reduced vertical stratification compared to the North Pacific, the surface signature of which being a much larger
surface density (+0.51 kg/m³), which provides favorable conditions for the emergence of deep convection. According
to the equation of state of seawater, the larger surface density found in the Weddell Sea convection region is due to
higher salinity (+0.73 psu) contributing to a density increase of 0.57 kg/m³, partly balanced by warmer temperature
(+0.36°C) contributing to a 0.06 kg/m³ density decrease.
Two major causes may explain these large differences in salinity between the two regions: *(i)* the atmospheric
circulation and freshwater fluxes to the ocean and *(ii)* the ocean circulation and positive salt-advection feedback. The
surface freshwater budget for two extended regions over the Weddell Sea (78°S–61°S and 62°W–8°E) and the North
Pacific (48°N–67°N and 124°E–143W) is shown in Table 3. Averaged precipitation and runoff over the box in the
North Pacific (respectively 1.48 and 0.37 m/yr) exceed those in the Weddell Sea (1.01 and 0.22 m/yr) by roughly 50%,
whereas the evaporation rates are almost similar (0.75 vs 0.71 m/yr). Overall, the average net surface freshwater input
is 0.53 m/yr in the Weddell Sea compared to 1.11 m/yr in the North Pacific. Reduced freshwater input by precipitations
and continental runoff is related to different atmospheric circulation in the Northern and Southern Hemispheres. In
the Southern Ocean, winds largely follow the Antarctic orography (as shown by geopotential height at 850 hPa, Fig.
S3), and induce almost no precipitation by orographic uplift over Antarctica coastal regions and no runoff to the
Southern Ocean. In contrast, in the North Pacific, the westerlies are blocked by the paleo-Rocky Mountains, especially
in the high-altitude region between 50°N and 70°N. The orographic uplift of moist air masses induces high
precipitation (up to 2–3 m/yr) and runoff into the North Pacific (as found in several other models, Carmichael et al.,
2016), leading to low sea surface salinity (below 30 psu) along the Pacific coast of North America (not shown) hence
increased surface stratification. The upper branch of the MOC and the associated poleward advection of saline
subtropical waters constitutes the other contribution to the larger salinities found in the Southern Ocean relative to the
North Pacific. This process was dumbed as a positive salt-advection feedback by Ferreira et al. (2018)
In contrast, present-day circulation, characterized by deep-water formation in the North Atlantic, is maintained by
higher salinities in the North Atlantic than in the North Pacific, which are partly sustained by atmospheric fluxes and
the salt-advection feedback (Ferreira et al., 2018). A recent sensitivity study of the impact of topography on modern
ocean circulation reveals that the presence of the Rocky Mountains influences the global salinity pattern and the
regions where deep convection occurs, through the adjustment of the freshwater transfers from the Pacific to the
Atlantic Oceans (Maffre et al., 2018).
**3.3. Factors contributing to the vigorous SOMOC**
Different factors contribute to the intense SOMOC simulated by the 55 Ma-3x simulation (40 Sv) in comparison with
typical present-day MOC, the intensity of which reaches around 18 Sv and 20 Sv for the upper and lower cells
respectively (Lumpkin and Speer, 2007). Deep-water formation occurs in the three sectors of the Southern Ocean
(Pacific, Atlantic and Indian) but the zonal connections between the different basins hamper a clear quantification of
the contribution of each deep-water formation sector to the SOMOC. However, the contributions of the Weddell Sea
and the Pacific sector of the Southern Ocean can be estimated because the narrow width of Drake and Tasman passages



at 55 Ma (Fig. 1) creates latitudinal continental boundaries on western and eastern sides of these regions. The Weddell
Sea, which exhibits the largest MLD in 55 Ma-3x, and the Pacific sector contribute roughly equally to the SOMOC
intensity (~ 19 Sv). The South Indian sector contribution is more difficult to assess directly because of the large open-
ocean zonal connection with the Atlantic sector.
The shallow Drake Passage at 55 Ma provides a western boundary for the development of a subpolar gyre in the
Weddell Sea (see Section 4 for more details). This clockwise gyre produces a favorable environment to trigger deep-
water formation (known as preconditioning) through isopycnals doming in the center of the gyre, thereby bringing
weakly stratified waters of the ocean interior close to the surface (Marshall and Schott, 1999). Clockwise subpolar
gyres, and associated deep-water formation by winter convection, are also present in the Pacific sector of the Southern
Ocean, in the Ross and Amundsen seas. Previous numerical investigations of the effects of a closed Drake Passage on
ocean dynamics have revealed that the closure of the Drake Passage tends to promote the existence of subpolar gyres
in the Southern Ocean and vigorous deep-water formation (Nong et al., 2000; Sijp and England, 2004; Ladant et al.,
2018). Additionally, it has been recently suggested that the effects of the closure and opening of the Drake Passage
and the Panama Seaway may not be independent (Yang et al., 2014; England et al., 2017; Ladant et al., 2018). For
instance, Yang et al. (2014) found that closing the Drake Passage tends to suppress the AMOC and to promote the
emergence of a strong SOMOC when the Panama gateway is open, whereas the AMOC may remain intense when the
Panama gateway is closed. It is thus very likely that, in our 55 Ma simulations, the very shallow Drake Passage,
together with the opened Panama gateway, both contribute to the strong SOMOC.
In addition to the influence of the different gateways, tidally-induced mixing, which represents the enhanced vertical
diffusivity resulting from the breaking internal waves generated by the interaction of tidal currents with rough bottom
topography (St. Laurent et al., 2002), is another factor that contributes to the strong SOMOC found in the 55 Ma-3x
simulation. A twin experiment of the 55 Ma-3x simulation, in which no tidal-induced mixing is prescribed (55 Ma-
3x-noM2), simulates a SOMOC with a similar structure but an intensity that is 7 Sv weaker (33 Sv, compared to 40
Sv in the reference 55 Ma-3x simulation, Fig. S4). It should be noted that the additional simulation has only been run
for 2000 years, so that a small part of the difference between the two runs could arise from different equilibrated states.
Yet, the difference between these two simulations is consistent with the recent results of Weber and Thomas (2017),
who find a 10 Sv MOC enhancement in an early Eocene simulation with the ECHAM5/MPIOM model that explicitly
simulates tides. This suggests that our parameterization for tidally-induced mixing based on the $M_2$ dissipation fields
of Green and Huber (2013) reasonably represents the effect of early Eocene tides on global ocean circulation. The
strengthening of the MOC induced by tidal mixing can be directly related to the driving role of diapycnal mixing on
the overturning circulation. Numerous studies have demonstrated that both the magnitude and the vertical distribution
of the diapycnal mixing largely affects the strength of the MOC (Bryan, 1987; Manabe and Stouffer, 1999). As already
pointed out by Green and Huber (2013), the Eocene MOC may be much more sensitive to the intensity of the abyssal
mixing than the present day AMOC, that is largely isolated from the ocean floor by the presence of AABW and
sustained quasi-adiabatically by the wind-driven upwelling in the Antarctic Circumpolar Current (ACC; Marshall and
Speer, 2012). Diapycnal mixing is the only process that can warm up the dense waters formed in the Eocene Southern



Ocean, and these dense waters are directly exposed to the tidally-induced mixing at the bottom, such that the abyssal
dissipation becomes the main controlling factor of the SOMOC intensity. In this respect, the large tidal dissipation
rate suggested by Green and Huber (2013) in the Pacific is particularly important.
Another often-mentioned factor affecting ocean circulation is the climate state, in response to the atmospheric $CO_2$.
Based on highly simplified models, theoretical studies have suggested that ocean ventilation tends to increase under a
warmer climate state due to a higher seawater density sensitivity to temperature (e.g. de Boer et al., 2007). However,
the deepwater formation is a very regional phenomenon and thus this idealized relation might be complicated by other
regional scale factors. Indeed, we did not see any systematic change in SOMOC between two $CO_2$ levels (1.5x and
3x) simulations (see Section 6 for further discussion).

### 4. Horizontal circulation and winds

In the present day North Atlantic, vertical and horizontal circulations are intimately connected, especially in the
subpolar gyre (Marshall and Schott, 1999). For instance, the warm North Atlantic Current flowing northward and the
cold East Greenland Current flowing southward are found roughly at the same depth at 60°N, such that the MOC in
z-coordinates does not capture the associated water mass transformation at high latitude, whereas the MOC in density
coordinates does (Zhang, 2010). It is only when the cold branch deepens in the Labrador Sea and becomes the Deep
Western Boundary Current off Cape Hatteras that the overturning streamfunction in z-coordinates provides a good
estimate of the water mass transformation. Hence, the horizontal subpolar gyre is really part of the North Atlantic
thermohaline circulation. By analogy, one would expect that similar connections exist in the Southern Ocean during
the Eocene, where the near-closure of the Drake and Tasmanian Passages allows for the emergence of intense subpolar
gyres, that will precondition and feed the formation of deep water sustaining a strong SOMOC. We thus examine in
the following the horizontal ocean circulation during the Eocene.

### 4.1. Gyre circulations

In the 55 Ma-3x run, several horizontal gyres are well developed in both hemispheres, as shown by the barotropic
streamfunction (Fig. 5). In particular, the closed Drake and Tasmanian passages support the western boundary currents
necessary for the buildup of intense subpolar gyres in each sub-basin of the Southern Ocean, with intensity of 40 Sv
in the Weddell Sea, 35 Sv in the Indian sector, and 28 Sv in the Ross and Amundsen Seas in the Pacific sector, and
winter convection and deep water formation is occurring in the center of these gyres, as it is the case in the modern
Labrador Sea (Marshall and Schott, 1999). The formation of deep water in the subpolar gyres is also promoted by the
advection of saline subtropical waters from the subtropical gyres southward flowing branch (visible for instance on
sea surface salinity), as shown by Ferreira et al. (2018). Compared to the PI conditions, the subtropical gyres are
strongly perturbed by the numerous open gateways connecting the different basins. In the Southern Hemisphere, due
to the large opening between Australia and Asia, the main subtropical gyre extends over both the paleo-Indian and
Pacific oceans, with a western boundary current leaning partly on the northern coast of Australia (up to 52 Sv),
Madagascar (60 Sv) and Africa. This 'super'-subtropical gyre is partly fed by a 31 Sv eastward flow south of the Cape



of Good Hope originating from the South Atlantic subtropical gyre. Fig. 5 also reveals the existence of a strong anti-
clockwise subtropical gyre south of a clockwise subpolar gyre in the North Pacific, with maxima of ~42 and 13 Sv
respectively. In contrast, the gyres in the tectonically-restricted North Atlantic basin (Fig. 1) are weak, with a
maximum of 22 Sv and 2 Sv for the subtropical and subpolar cells respectively. The numerous gateways in the tropical
band clearly complicate the traditional gyre pattern in each basin, and increase their global connectivity.
In the PI-1x simulation, the maximum of the streamfunction in the North Atlantic subtropical and subpolar gyres are
respectively 37 and 19 Sv, which is comparable with the intensity of the gyres found in the Southern Hemisphere in
the 55 Ma simulations. The most salient feature of the PI-1x simulation is the existence of the Antarctic Circumpolar
Current (ACC), with an eastward transport of 108 Sv through the Drake Passage, which totally disrupts the subtropical-
subpolar gyres circulation in the different basins of the Southern Hemisphere. In many respects, the South Atlantic
and the Weddell Sea during the Eocene are thus an analogue of the present-day North Atlantic in terms of subtropical,
subpolar and overturning connections.
**4.2. Wind stress**
The modern ocean circulation, in particular in the surface layers, is largely driven by the surface winds (e.g. Munk,
1950), and it is thus interesting to examine the difference between the PI and 55 Ma horizontal circulation in light of
changes in the wind pattern.
Overall, the patterns of the wind stress at the ocean surface in the 55 Ma-3x and PI-1x simulations are similar (Fig.
6), although the magnitude differs between the simulations, and the sign of the difference depends on the hemisphere
considered. Indeed, the wind stress is about 30% weaker in 55 Ma-3x than in PI-1x in the Southern Hemisphere (by
0.05 N m$^{-2}$ at 45°S) whereas it is slightly stronger in the Northern Hemisphere. In the Southern Hemisphere, the
difference is particularly striking in the South Atlantic and Indian basins and is largely due to the blocking position of
Australia in the early Eocene. The paleogeographic context also explains the increased symmetry in zonal wind stress
fields between the Eocene hemispheres relative to the PI.
The more symmetrical pattern of wind stress at the ocean surface in the 55 Ma-3x simulation (compared to PI-1x) is
found in the zonal wind fields from the surface to 500 hPa in the atmosphere (Fig. 6D). The zonal wind strength is
largely determined by the meridional temperature gradient in the atmosphere through the thermal wind relation
(Holton and Staley, 1973). Indeed, the meridional temperature gradient in the 55 Ma-3x run (compared to PI-1x) is
much reduced in the Southern Hemisphere south of 40°S in a large part of the air column, from the surface up to at
least 500 hPa, in good agreement with the weaker westerly winds found in the 55 Ma-3x run (Fig. 6D). Moreover, the
positions of Australia, Africa, and South America, all much more south during the Eocene than now, result in a
blocking effect on the zonal winds, reducing the wind at 500 hPa by a maximum of 18% at 40°S. In the Northern
Hemisphere, the meridional temperature gradient in the 55 Ma-3x run is reduced at the surface only north of 60°N,
and similar to the PI-1x run at 500 hPa, whereas the zonal winds are slightly stronger at 55 Ma throughout the air





column. At the surface, the maximum zonal wind stress are 19% stronger and shifted poleward by 2°, mainly due to
land-sea distribution.
Changes in the ocean gyres circulation between 55 Ma and PI configurations are mostly due to the large changes in
the ocean basin geometry and gateways, and not to the moderate changes in the strength and patterns of the wind stress
(and its curl). A full understanding of what sets the intensity of the gyres in the 55 Ma simulations (as well as in the
PI simulations) would require further investigations, which are beyond the scope of the present paper.
**5. Oceanic heat transport and its decomposition**
The modern ocean circulation plays a key role in the regulation of the climate through its contribution to the
redistribution of heat from the Equator to the Poles. In the tropics, the ocean transports roughly 50% of the 3 PW
carried northward by the ocean–atmosphere system, but less than 10% of the total at high latitude (Trenberth and
Caron, 2001). The relative contributions of the horizontal and overturning ocean circulations to the meridional heat
transport also vary greatly over different latitudes and between oceanic basins (Ganachaud and Wunsch, 2003). In
light of the different ocean circulations found in the Eocene and PI conditions, we further investigate in the following
how efficient was the ocean at transporting heat across latitudes in the early Eocene.
**5.1. Oceanic heat transport**
The total meridional OHT at a given latitude y is defined as the sum of advective and diffusive contributions:
$$OHT_{total} = \rho_0 C_p \int_W^E \int_{-H}^0 \left( v\theta + K_H \frac{d\theta}{dy} \right) dz dx \qquad (2)$$
where $\rho_0$ is the seawater density, $C_p$ is the specific heat capacity of sea water, v is the meridional velocity, θ is the
potential temperature, $K_H$ is the horizontal diffusivity coefficient, H is the ocean depth. Here the computation is
performed from the model output at each model time step. For the 55 Ma simulation, there is a significant OHT
increase in the Southern Hemisphere compared to PI simulations, but a decrease in the Northern Hemisphere (Fig. 7).
As a result, the simulated OHT in the 55 Ma-3x experiment is remarkably asymmetric between hemispheres. The
mean OHT difference between the 55 Ma-3x and PI-1x simulations is of the order of 0.2 PW (1 PW = $10^{15}$ Watt),
peaking at 0.5 PW around 35°S. The 55 Ma-3x OHT reaches a maximum of 1.7 PW at 15–20°S, that is ~0.3 PW
larger than in the PI simulations at the same latitude, but also ~0.5 PW larger than the maximum PI value in the
Northern Hemisphere. This larger OHT in the Southern Ocean contributes to maintain the Southern Hemisphere
particularly warm in the Eocene, especially south of 50°S, as can be seen on the SST distribution (Fig. 2).
Previous studies have examined the role of OHT during the Eocene with a particular focus on the response of the OHT
to the opening of Southern Ocean gateways in either true Eocene paleogeography or more idealized modern
configurations. The OHT simulated by our 55 Ma-3x experiment lies within the range of values found in the literature.
For instance, using the NCAR CCSM ocean model with surface heat flux boundary conditions mimicking an energy-





balanced atmospheric model, Nong et al. (2000) found that closing the Drake Passage in a modern configuration
results in a stronger SOMOC (24 vs 12 Sv), associated with an increased poleward OHT in the Southern Hemisphere
(+0.2 PW, from 1 to 1.2 PW) and a decreased OHT in the Northern Hemisphere. Similarly, using the UVic
intermediate complexity Earth System Climate Model, Sijp and England (2004) found a strongly enhanced heat
transport (from 1.6 to 2.4 PW) in the Southern Hemisphere in response to the closure of Drake Passage in a modern
configuration. Using the fully coupled NCAR model with a closed Drake Passage but a closed Tasmanian gateway in
a realistic Eocene configuration, Huber et al. (2004) found a rather weak poleward OHT in the Southern Hemisphere
during the Eocene (with a maximum of 0.9 PW at ~10ºS), likely because of the absence of a strong SOMOC in their
simulations compared to ours. When closing the Drake passage in the GFDL model, Yang et al. (2014) found that the
change in OHT was much larger when the Panama Seaway was open, with a strong increase of the OHT in the
Southern Hemisphere. More recently, Baatsen et al. (2018) used the higher resolution CESM coupled model to
simulate the 38 Ma climate, prescribing levels of $CO_2$ and $CH_4$ in the atmosphere 2 and 4 times the PI levels. In these
two simulations, they found a maximum of ~1.5 PW OHT at 20ºS in the Southern Hemisphere, associated with a 14–
16 Sv SOMOC (compared to the 40 Sv and 1.7 PW at the same latitudes in our 55 Ma-3x simulation). Although these
previous studies are all based on results from different models of various complexity and resolution, they all
consistently suggest that the OHT is largely coupled to the structure and strength of the MOC cells. For instance, the
difference in the MOC intensity simulated by coupled and uncoupled models could be primarily caused by the positive
salt-advection feedback and the self-stabilizing thermal feedback (Sijp and England, 2004). It is noteworthy that the
strengthening influence of tidal-induced mixing on the MOC (+7 Sv) is associated with a rather weak increase in
OHT, lower than 0.03 PW on average, in agreement with Weber and Thomas (2017). Such a nonlinear relationship
between OHT and MOC has also been suggested by Boccaletti (2005), who stressed that, locally, the shallow
circulation can be as important as the deep overturning for determining the OHT.

**5.2. Decomposition of the meridional ocean heat transport**

In order to understand the differences in OHT among our simulations (Fig. 7A), we further split up the OHT into an
advective contribution (OHT$_{adv}$) and a diffusive contribution, corresponding respectively to the first and second term
on the RHS of Eq. 2 (Fig. 7B and 7C). This decomposition reveals that, in both the 55 Ma and the PI runs, the advective
part dominates the OHT at all latitudes, except at 40°S/N in PI where the presence of large temperature gradient (Fig.
2B) results in larger diffusive heat transports. Figs. 7B and 7C also reveals that different OHT between PI and 55 Ma
are mainly due to differences in the advective components, the differences in diffusive OHT being rather small.
The advective OHT$_{adv}$ can be decomposed further into an overturning (OHT$_{MOC}$) and a gyre (OHT$_{gyre}$) component,
following for instance Bryan (1982) or Volkov et al. (2010):

$$OHT_{adv} = \rho_0 C_p \iint \bar{v}\,\bar{\theta}\,dx\,dz + \rho_0 C_p \iint v'\,\theta'\,dx\,dz \qquad (3)$$

where $\bar{v},\ \bar{\theta}$ represent the zonal averages of the velocity and temperature, respectively, and $v'$, and $\theta'$ the deviations
from these zonal means. The first term of the RHS of Eq. 3 corresponds to the overturning component (OHT$_{MOC}$) and



the second term corresponds to the horizontal transport associated with the large-scale gyre circulation (OHT$_{gyre}$).
Note that due to limitations on the availability of model outputs, the different terms of Eq. 3 presented on Figs. 7D
and 7E are computed from monthly means. This explains why the sum of the two terms does not completely equal
OHT$_{adv}$ shown on Fig. 7C, the latter being computed at each model time step during the simulations. The differences
between the two computations can be seen on Fig. 7C.
The decomposition reveals that the enhanced Southern Hemisphere OHT$_{adv}$ at 55 Ma (compared to PI) is overall due
to differences in OHT$_{MOC}$ (Fig. 7D, 7E). The contribution from the gyre circulation varies with latitude, with a
compensation effect between OHT$_{gyre}$ and OHT$_{MOC}$ in the low-latitudes, and an enhancement in mid-to-high latitudes.
Consequently, in the tropics, the strong Eocene OHT$_{MOC}$ (up to 2 PW) is ~0.3 PW larger than in the PI simulations,
leading to an overall larger OHT$_{adv}$ in the 55 Ma simulations. In the mid-latitudes of the Southern Hemisphere, where
the OHT is overall smaller than in the tropics, the OHT$_{MOC}$ is almost 1 PW stronger than in PI. By contrast, south of
60°S, enhanced OHT in 55 Ma simulations (compared to PI) results from a combination of stronger OHT$_{gyre}$ and
OHT$_{MOC}$. It is worth noticing that OHT$_{gyre}$ at ~40°S in the PI-1x simulation is very likely underestimated in our
decomposition computed from monthly mean data, because higher-frequency processes (resulting for instance from
atmospheric synoptic variability) could contribute significantly to OHT$_{gyre}$ in regions where the mean meridional
currents are weak (Volkov et al., 2010).
It is obvious from Figs. 3 and 7 that the vigorous SOMOC simulated in the 55 Ma experiment drives a strong net OHT
toward the South Pole. This strong SOMOC is associated with a poleward transport of warm waters at shallow depths
where zonal oceanic temperature gradients are larger, and a returning equatorward transport of colder water at depth
where ocean temperature tends to be more homogenous (Fig. 2C). Remarkably, although the ACC is absent from the
Eocene simulation because of different Drake and Tasmanian passages configurations that constitute latitudinal
barriers (Munday et al., 2015), the contribution of the gyre circulation and diffusive process to poleward heat transport
(OHT$_{gyre}$) is smaller than in PI simulations. This small OHT$_{gyre}$ in the 55 Ma simulation is unexpected, given previous
hypotheses on the climatic effects of the ACC (e.g. Nong et al., 2000; Toggweiler and Bjornsson, 2000; Sijp and
England, 2004). The ACC has indeed been suggested to be a barrier for poleward heat transport, so that the onset of
the ACC could be a potential driver for the Eocene-Oligocene Antarctica cooling around 34 Ma. Yet, these studies
may not have captured the full complexity of the links between the ACC and the OHT in the Southern Ocean. Indeed,
the analysis of both in-situ observations (Watts et al., 2016) and the CESM1.0 model (Yang et al., 2015) have revealed
that the ACC is composed of meridional excursions of the mean geostrophic horizontal shear flow, energetic eddies
and large diffusive heat transport, which balances out the equatorward OHT due to Ekman transport and leads to a net
poleward OHT in the Southern Ocean (Volkov et al., 2010).
**6. Sensitivity of the ocean response to a doubling of the levels of atmospheric CO$_2$**
Our analysis has so far focused on the comparison between the 55 Ma-3x and the PI-1x simulations, as the former is
performed with atmospheric CO$_2$ levels thought to be representative of the early Eocene (Foster et al., 2017). The



analysis of two additional simulations (55 Ma-1.5x and PI-2x) allows us to investigate the robustness of the ocean circulation in this range of atmospheric $CO_2$ concentration. The set of simulations based on both the 55 Ma and PI configurations also help us to quantify the sensitivity of the oceanic conditions to a doubling of the level of $CO_2$ in the atmosphere for early Eocene and modern setting.

The mean ocean temperatures are very sensitive to the atmospheric $CO_2$ concentration in the Eocene configuration (Table 1). Global mean ocean temperature in 55 Ma simulations increases by 4.9°C in response to $CO_2$ doubling from 1.5x to 3x, which is much larger than the 1°C increase in PI simulations from 1x to 2x. SST also shows a much larger increase at 55 Ma (+4.7°C) than in PI simulations (+2.6°C), especially at high latitudes and in the regions of deep-convection of the Southern Ocean (Fig. 8), which is similar to the changes in air temperature at 2 m (+5.6°C vs +3.5°C respectively, this difference is known as the climate sensitivity). Such contrasted values of climate sensitivity between 55 Ma and PI are in good agreement with the recent results of Farnsworth et al. (2019) when analyzing a series of climate models. In the absence of sea-ice at 55 Ma, the winter SST in deep-convection regions is largely influenced by the air-sea interactions and thus directly related to air temperature, that rarely decreases below 10°C (resp. 5°C) in 55 Ma-3x (resp. 1.5x), such that the deep waters filling the whole ocean vary accordingly with temperature. This is not the case in the present-day configuration where deep-water formation is tied to the marginal ice zones, such that the dense water formed through the effect of the brine rejection have initial temperatures close to freezing. Intuitively, we could have expected a larger sensitivity of ocean temperature to the level of $CO_2$ in the atmosphere in the PI runs, induced by the ice-albedo feedback. Yet, the effect of this feedback appears to be limited to the high latitudes (Fig. 8), and only plays a marginal role for the changes in global mean SSTs or temperatures of the deeper water masses (Table 1).

The ocean circulation only shows a minor response to a $CO_2$ doubling in both the 55 Ma and PI configurations, although a regional response still exists (Table 4). In the 55 Ma simulations, doubling $CO_2$ enhances the maximum abyssal SOMOC by only 0.3 Sv out of 40 Sv in total, while the intensity of the shallow MOC cell and of the barotropic streamfunction are slightly reduced. In the PI simulations, doubling the level of atmospheric $CO_2$ has the opposite effect on the MOC cell occupied by the AABW, whose maximum intensity reduces by 1.2 Sv (out of ~15 Sv), whilst the AMOC slightly increases by 0.3 Sv (out of ~11 Sv). This small increase in the steady-state AMOC is quite interesting and needs to be contrasted with the transient response of the AMOC intensity to global warming (Gent, 2018; Jansen et al., 2018). Indeed, CMIP-type climate models consistently project a strong decline of the AMOC strength when forced with a range of increasing greenhouse gas emission scenarios (Schmittner et al., 2005; Cheng et al., 2013). Yet, when run for longer integrations until full equilibrium, models suggest that the AMOC tends to recover, so that the AMOC is not very sensitive to the level of $CO_2$ (Jansen et al., 2018; Thomas and Fedorov, 2019), as is the case in our PI simulations. Once equilibrium is reached, the most significant effect of a doubling in the $CO_2$ concentration in our experiments is a sharp increase in the ACC transport (+22 Sv out of 108 Sv in PI-1x), in response to stronger westerlies in the Southern Ocean. In contrast, the barotropic circulation remains almost the same in the Pacific, but varies in the North Atlantic with a 5 Sv stronger (weaker) subpolar (subtropical) gyre in the PI-2x simulation compared to PI-1x.





A number of paleoclimate studies have investigated the influence of $CO_2$ levels on ocean circulation in coupled
models, with contrasted responses depending on the period considered and the model used. For instance, the deep
overturning circulation in the HadCM3L climate model shows an overall high sensitivity to $CO_2$ concentrations in
simulation of the Paleocene-Eocene period (Lunt et al., 2010), whilst the GENIE model only exhibits a small response
in the Cretaceous simulations of Monteiro et al. (2012). Winguth et al. (2010) further suggest that, in a given simulation
of the Paleocene-Eocene period, the different MOC cells (e.g. in the Northern and Southern hemispheres) could
respond differently to a change of $CO_2$ levels. These various responses can be attributed to different factors. First, it
is clear that the resolution and overall complexity of the model used for these studies may partly control the sensitivity
of the MOC to $CO_2$ levels, as in models of modern climate (e.g. Bryan et al., 2006). Second, a variety of time scales
are intertwined in the adjustment of the ocean circulation to external perturbations, from decades for the dynamical
adjustment to millennia for the thermo-dynamical response of bottom waters through vertical advective-diffusive
balance (e.g. Donnadieu et al., 2016). The transient response of the ocean can therefore differ, or even be in the
opposite direction, from the final equilibrium response.
The OHT response to a doubling of $CO_2$ in the Eocene simulations is also rather small, with a slight decrease, in
contrast to the OHT increase seen in the PI simulations (although the magnitude of the change is smaller; Fig. 9A).
The OHT in the 55 Ma-3x simulation is about 0.15 PW smaller than in 55 Ma-1.5x simulation over most of latitudes.
Both overturning and horizontal components contribute to this overall smaller OHT in the 55 Ma-3x simulation (Fig.
9D, 9E). A weak OHT in the tropics is due to weaker MOC at the same latitudes, while a weak OHT at high-latitude
can be attributed to the small amplitude of horizontal gyres. For the PI simulations, the OHT in the PI-2x simulation
is ~0.1 PW larger than in PI-1x at high latitudes. This larger OHT in PI-2x is mostly due to the gyre component, which
is in good agreement with the ~5 Sv stronger North Atlantic subpolar gyre and almost no change in the AMOC for
instance.
Our results therefore support a stable, yet rather small, response of global ocean circulation and heat transport to the
doubling of atmospheric $CO_2$ levels. Nevertheless, we only investigate a limited range of $CO_2$ levels (from 1.5x to 3x)
and cannot exclude that the sensitivity of ocean circulation to $CO_2$ concentrations may change at more extreme $CO_2$
levels, as the response of the ocean conditions is highly non-linear (Lunt et al., 2010). Given that the levels of $CO_2$ in
the Eocene atmosphere are relatively poorly constrained by proxy reconstructions, additional experiments are
underway to explore higher values of $CO_2$ concentration for the early Eocene configuration.
**7. Conclusions**
The early Eocene (~55 Ma) was most likely the warmest period in the Cenozoic. During that period, paleogeographic
restrictions of certain modern basins and gateways, such as the North Atlantic and the Drake and Tasmanian passages,
suggest fundamental differences with the modern large-scale ocean circulation. It has been proposed that the distinct
mode of ocean circulation operating during the early Eocene may have contributed to the significant polar warmth
recorded by observational evidence. There is however no consensus on the modes of early Eocene ocean circulation



or on the relative influence of the overturning and horizontal circulation on the poleward heat transport. Here we revisit this question by analyzing the ocean circulation and its contribution to the meridional OHT using simulations of the Early Eocene performed with the IPSL-CM5A2 coupled climate model set up with the recent paleogeographic reconstructions of this Eocene time slice distributed as part of the DeepMIP project. Our main results are summarized hereafter.

A strong abyssal overturning circulation is found in the 55 Ma simulation, with deep water formed only in the Southern Ocean (mainly in the Weddell Sea), whereas there is no deep water formation in the Northern Hemisphere, in contrast to some previous work on the early Eocene (e.g. Winguth et al., 2012). This situation is favored by orographically-induced freshwater fluxes (precipitation and runoff) and maintained by a salt-advection feedback. Indeed, the atmospheric circulation around Antarctica induces relatively low precipitation rates in the Southern Ocean, resulting in higher salinity, and hence larger surface density than in the North Pacific, where the large precipitations and runoff induced by the orographic uplift of the westerlies above the paleo-Rockies tend to reduce the surface salinity and inhibit deep water formation.

The paleogeography (and paleobathymetry) and tidally-induced mixing are the main drivers of the strong SOMOC (up to 40 Sv) during the early Eocene. The (nearly-)closed Drake and Tasmanian Passages are of fundamental importance for sustaining the SOMOC, via their effect on the horizontal ocean circulation. More specifically, with the (nearly-)closed Drake and Tasmanian Passages serving as a western boundary, clockwise subpolar gyres are well-developed (~40 Sv) in the Weddell and Ross Seas, favoring the emergence of deep convection and deep-water formation through isopycnal doming and salt-advection feedback. Tidal-induced mixing also contributes to 7 Sv (out of 40 Sv) to this SOMOC, but with only a limited impact on the heat transport.

The vigorous SOMOC simulated for the Eocene is associated with a larger poleward heat transport (by a maximum of 0.5 PW relative to PI) in the Southern Hemisphere, that largely contributes to maintain a warm Southern Ocean and Antarctica. Perturbation experiments have been conducted in present-day coupled models to evaluate the impact of an AMOC shutdown. In their model, Vellinga and Wood (2008) found that a 10 Sv reduction in AMOC, associated with a change of its structure, leads to a 1.7°C cooling of the Northern Hemisphere, with a local stronger cooling by 5°C in the northern North Atlantic. This gives credit to the importance of the 40 Sv SOMOC for maintaining the Southern Ocean warm in the 55 Ma simulations. However, other factors than a strong SOMOC and associated OHT could contribute to the warm Southern Ocean. Indeed, Rose and Ferreira (2013) have shown that changes in OHT can induce changes in global mean temperature and meridional temperature gradient through convective adjustment of the extratropical troposphere and increased greenhouse effect. According to their results, the magnitude of those changes could be up to 1°C and 2.6°C for every 0.5 PW enhancement in OHT. Further investigations would be required to examine if this mechanism is also at play in our simulation, and contributes significantly to the Southern Ocean warmth.

A further decomposition of the OHT reveals that the different overturning circulations between 55 Ma and PI explain most of the increase of the OHT in the Southern Hemisphere. The contribution of gyre circulation to the OHT is only





secondary, and varies with latitude, with a compensating effect between the MOC and gyre circulations in low- and mid-latitudes, whereas the two contributions add up in high latitudes. More importantly, the latitudinal distribution of the gyre contribution to the OHT only marginally varies between the 55 Ma and PI simulations, despite the absence of an ACC in the Eocene experiment. Given that the 55 Ma paleo-bathymetry does not allow the existence of a strong ACC, this questions strongly the idea that the ACC could be a strong barrier for the OHT in our modern climate, and suggests that the meridional excursions of the ACC might indeed play an important role in the gyre-related OHT (Volkov et al., 2010; Watts et al., 2016; Yang et al., 2015).

**Acknowledgements.** This research has received partial funding from the French National Research Agency (ANR) under the 'Programme d'Investissements d'Avenir' ISblue (ANR-17-EURE-0015) and LabexMER (ANR-10-LABX-19) for the COPS project. Additional funding has been received from Ifremer and 'Université Bretagne Loire' to support the postdoc of Yurui Zhang. Jean-Baptiste Ladant received funding from ANR during the early phases of this work. We thank the GENCI TGCC at CEA for providing HPC computational resources. We are grateful to Pierre Sepulchre for the initial design of the numerical model. We thank Arnaud Caubel, Anne Cozic, Agnès Ducharne, Josefine Ghattas, François Lott, Olivier Marti, Jean-Yves Peterschmitt and Alistair Sellar for their help in implementing the Early Eocene boundary conditions and in resolving technical issues.

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





## Tables

**Table 1** Summary of the simulation setup and key diagnostics in the different simulations used in this study. All the values presented are averaged over the last 100 years of each simulation.

| Simulation | Setup | | | Ocean surface | | | Full depth ocean | | | Atmos. |
|---|---|---|---|---|---|---|---|---|---|---|
| | $CO_2$ (ppmv) | Bathymetry | Duration (yr) | SST (°C) | SSS (psu) | Sigma (kg/m³) | T (°C) | S (psu) | Sigma (kg/m³) | T at 2m (°C) |
| 55 Ma-3x | 840 | 55Ma | 4000 | 27.51 | 34.04 | 21.33 | 11.30 | 34.68 | 26.24 | 25.12 |
| 55 Ma-3x-noM2 | 840 | 55Ma | 2000 | 27.47 | 34.16 | 21.44 | 10.85 | 34.68 | 26.34 | 24.98 |
| 55 Ma-1.5x | 420 | 55Ma | 4000 | 22.82 | 34.36 | 22.94 | 6.38 | 34.68 | 26.94 | 19.56 |
| PI-2x | 560 | PI | 2910 | 20.14 | 34.40 | 23.63 | 4.32 | 34.61 | 27.27 | 16.88 |
| PI-1x | 280 | PI | 2790 | 17.51 | 34.43 | 24.32 | 3.34 | 34.61 | 27.40 | 13.33 |

**Table 2** Root-mean-square-deviation (RMSD) of simulated annual mean SST and proxy-based SST estimates (in °C). RMSD metrics are defined in Supplementary material. The proxy-based SST estimates are from the DeepMIP dataset for the early Eocene (Hollis et al. 2019), and the number of data points and the uncertainty for each proxy type is also indicated. The uncertainty range is defined as the 2σ deviations for δ¹⁸O, and as the range between 5% and 95% percentile SST estimates for TEX⁸⁶, Mg/Ca and Clumped isotope data.

| Type of proxy | | TEX⁸⁶ | δ¹⁸O | Mg/Ca | Clum. isotope |
|---|---|---|---|---|---|
| Number of data points | | 10 | 10 | 7 | 5 |
| Uncertainty range of proxy-data | | 15.1 | 3.4 | 6.7 | 5.1 |
| RMSD | 55Ma-3x | 13.7 | 7.5 | 6.4 | 5.3 |
| | 55Ma-1.5x | 18.2 | 7.5 | 10.6 | 8.1 |

**Table 3** Key ocean surface parameters in the North Pacific and Weddell Sea in 55Ma-3x simulation. The Weddell Sea and the North Pacific convection regions are defined by the deepest mixed layer depth at high-latitudes and the extended regions are defined as boxes over the Weddell Sea (78°S-61°S and 62°W-8°E) and the North Pacific (48°N-67°N and 124°E-143°W) roughly corresponding to closed contours of sea surface height. SST, SSS, sigma are winter average (January-February-March for the Northern Hemisphere and July-August-September for the Southern Hemisphere), MLD is given for the end of winter (March for the Northern Hemisphere and September for the Southern Hemisphere), while precipitation, runoff and evaporation are annual means.

| | Convection region | | | | Extended region | | | | | | |
|---|---|---|---|---|---|---|---|---|---|---|---|
| | SST (°C) | SSS (psu) | sigma (kg/m³) | MLD (m) | Area (km²) | Precipitation (P) (m/yr) | Runoff (R) (m/yr) | Evaporation (E) (m/yr) | P+R-E (m/yr) | SST (°C) | SSS (psu) |
| North Pacific | 9.21 | 34.06 | 26.22 | 111 | 9.23e6 | 1.48 | 0.37 | 0.74 | 1.12 | 10.33 | 32.89 |
| Weddell Sea | 9.57 | 34.79 | 26.73 | 3129 | 4.70e6 | 1.01 | 0.22 | 0.69 | 0.54 | 10.13 | 34.77 |

**Table 4** Intensity of the gyres (SPG$_{SA}$: South Atlantic subpolar gyre for 55 Ma; STG$_{NA}$: North Atlantic subtropical gyre; SPG$_{NA}$: North Atlantic subpolar gyre; STG$_{NP}$: North Pacific subtropical gyre; SPG$_{NP}$: North Pacific subpolar gyre; ACC: Antarctic circumpolar current for PI) and overturning cells (SOMOC: Southern Ocean MOC for 55Ma; AABW: Antarctic bottom water for PI; NADW: North Atlantic deep water for PI).

| Simulation | Overturning (Sv) | | Gyre intensity (Sv) | | | | | | |
|---|---|---|---|---|---|---|---|---|---|
| | SOMOC/ AABW | NADW | SPG$_{SA}$/ ACC | SPG$_{SI}$ | SPG$_{SP}$ | STG$_{NA}$ | SPG$_{NA}$ | STG$_{NP}$ | SPG$_{NP}$ |
| 55 Ma-3x | -40 | — | 40 | 35 | 28 | 22 | -2 | 42 | -13 |
| 55 Ma-1.5x | -40 | — | 38 | 46 | 35 | 29 | 0 | 53 | -8 |
| PI-2x | -15 | 11.6 | 130 | — | — | 32 | -24 | 48 | -19 |
| PI-1x | -16 | 11.3 | 108 | — | — | 37 | -19 | 48 | -20 |



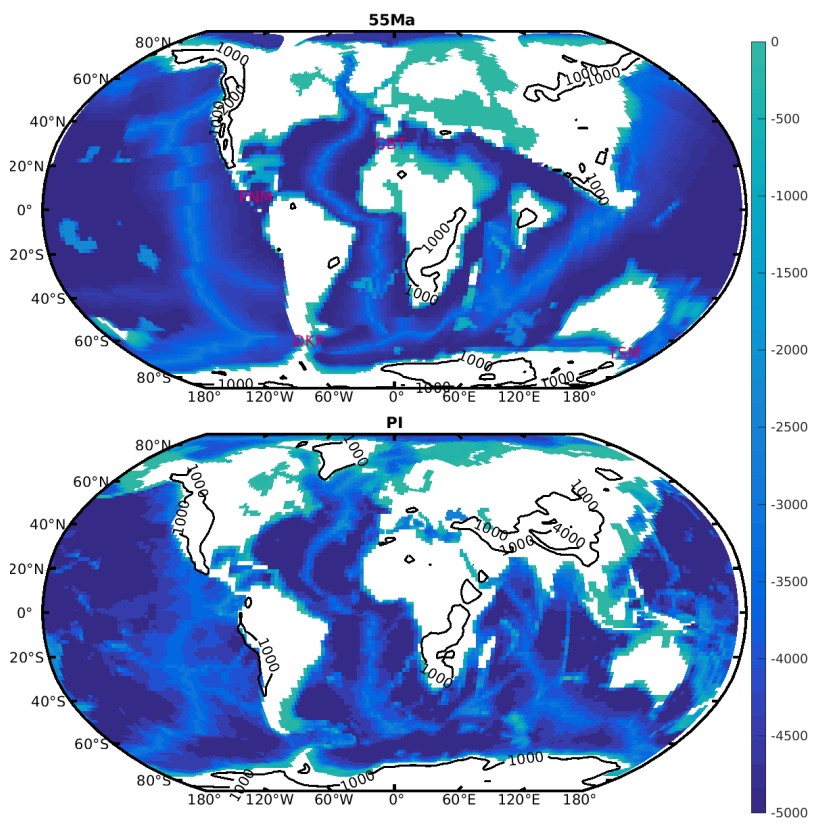

**Figure 1.** Bathymetry/topography (m) boundary conditions used in the 55 Ma (based on Herold et al. 2014) and PI simulations. The black contours indicate the 1000 and the 4000m altitude. DKP indicates the Drake Passage, TSM the Tasmanian Passage, PNM the Panama passage and GBT Gibraltar.

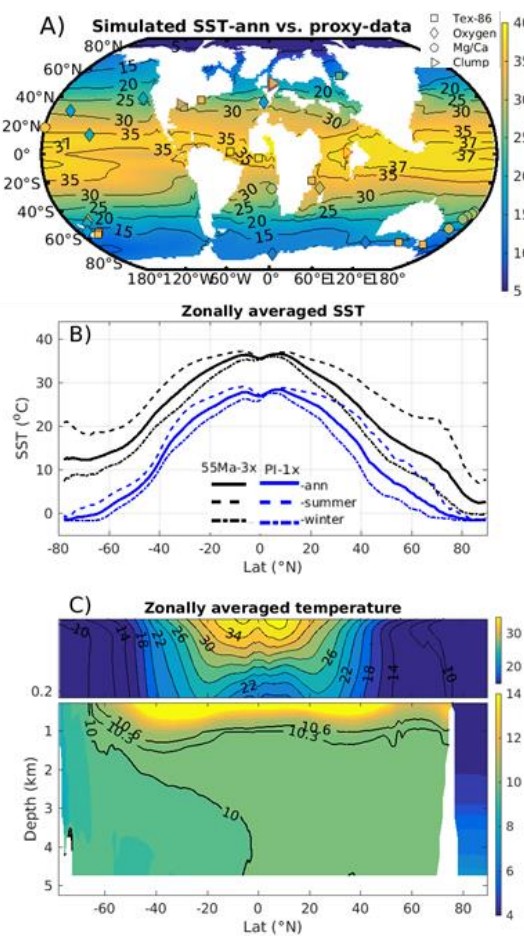

**Figure 2.** (A) Annual mean SST (in ºC) in the 55 Ma-3x simulation, and point-to-point comparisons with proxy-based SST estimates from the DeepMIP dataset for the early Eocene (Hollis et al. 2019). The different symbols represent different proxies. B) Simulated zonally-averaged annual mean SST (in ºC, solid lines) in the 55 Ma-3x (black) and PI-1x (blue) simulations. The dashed and dotted lines are indicating the means for Summer and Winter, respectively. (C) Zonally-averaged ocean temperature in the 55 Ma-3x simulation, with a zoom in the upper 200 m (contour interval: 2ºC and 0.3ºC for the top and bottom panels, respectively).



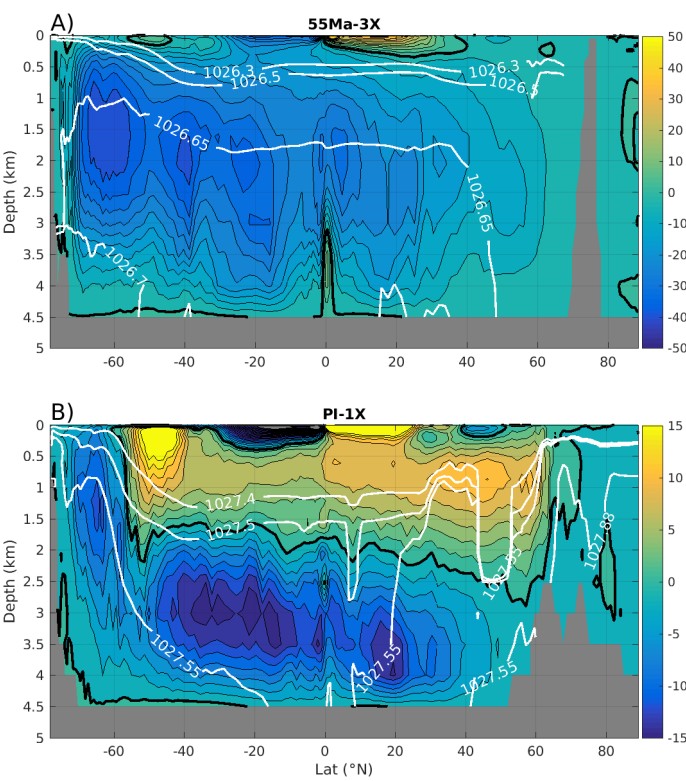

**Figure 3.** Streamfunction of the meridional overturning circulation (in Sv) in the 55 Ma-3x (A, contour interval: 4 Sv) and PI-1x (B, contour interval: 2 Sv) simulations. Note the different colorbars of the two subplots. The black thick lines indicate the zero-contour, with positive values indicating clockwise circulation, and negative values anti-clockwise circulation. White lines show selected zonally-averaged isopycnal contours (potential density in kg/m$^3$).



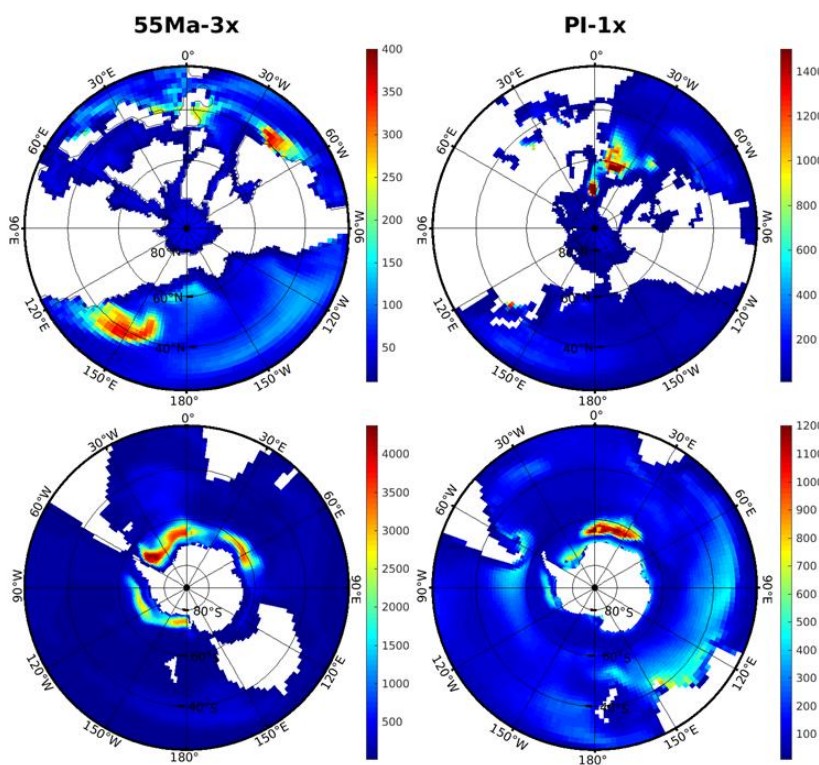

**Figure 4.** Winter (i.e. March in the Northern Hemisphere – top panels; and September in the Southern Hemisphere – bottom panels) mixed layer depth (in m) in the 55 Ma-3x (left) and PI-1x (right) simulations. Note the very different colorbars among plots. Mixed layer depth is defined by the potential density difference of 0.3 kg/m$^3$ with reference to the surface.




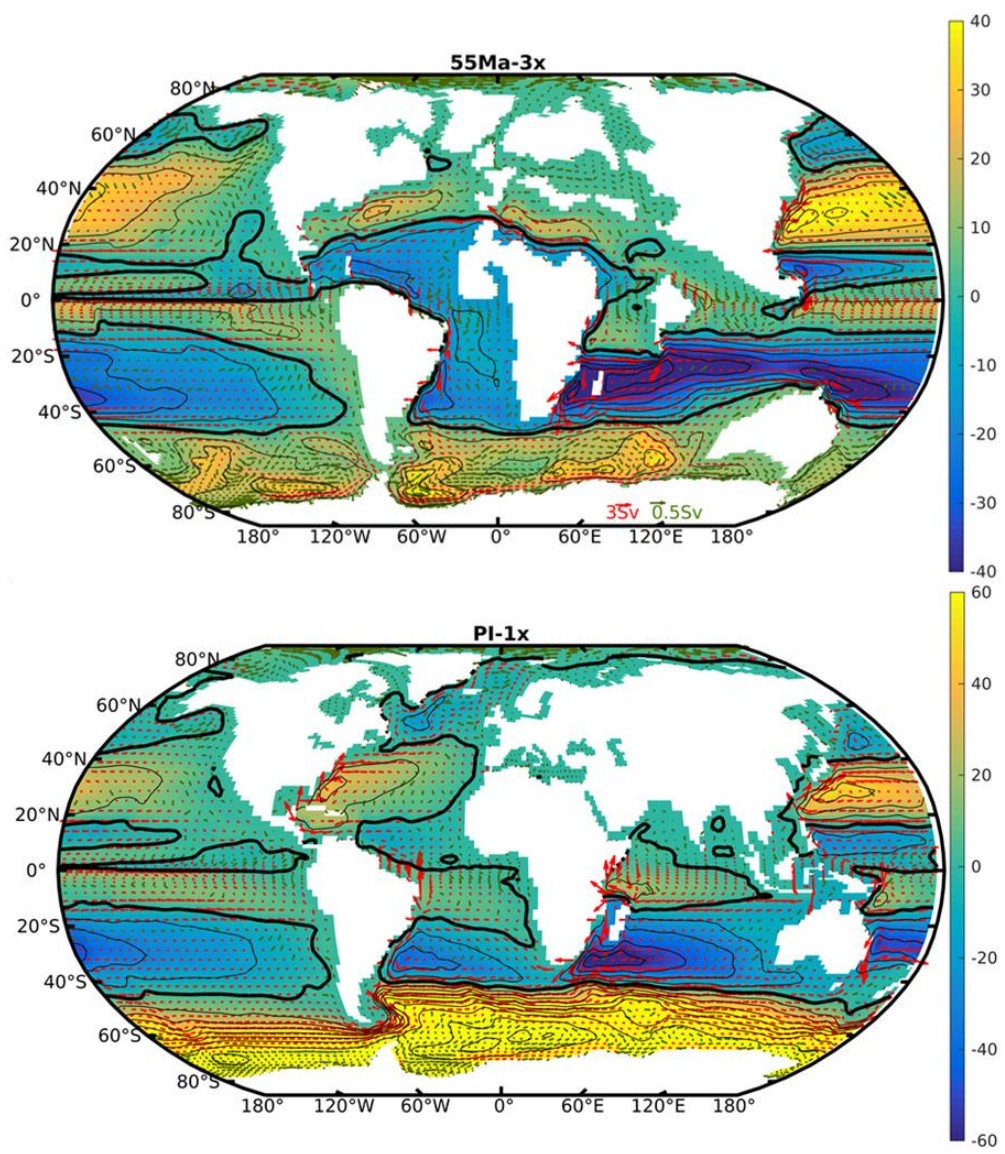

**Figure 5.** Barotropic streamfunction (in Sv) in the 55 Ma-3x (top, contour interval: 10 Sv) and PI-1x (bottom, contour interval: 15 Sv) simulations, integrated northward from Antarctica. Note the different colorbars of the two subplots. The black thick lines indicate the zero-contour. The mean transport (in Sv) integrated over the top 300m is indicated with vectors (note that only one every two point is plotted to increase the readability). Two scales are used to represent transport larger (in red) or lower (in green) than 0.5 Sv. The transports through the key gateways in the 55 Ma-3x simulation are respectively: Drake Passage 3.1 Sv, Tasmania 1.3 Sv, Panama 4.7 Sv, Gibraltar -14.3 Sv (positive transports are eastward, negative westward). The Drake Passage throughflow, corresponding to ACC, in the PI-1X simulation is ~108 Sv.





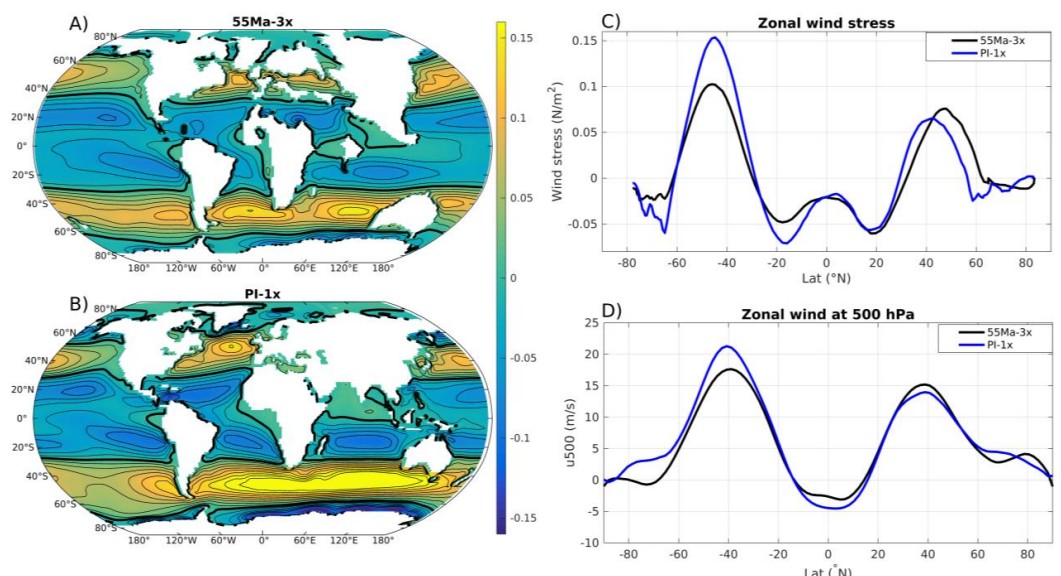

**Figure 6.** Zonal wind stress (in N/m²) in the 55 Ma-3x (A) and the PI-1x simulation (B). (C) Zonally averaged zonal wind stress (in N/m²) as a function of latitude in the 55 Ma-3x and PI-1x simulations. (D) Zonally averaged zonal wind (in m/s) at 500 hPa in the atmosphere as a function of latitude in the 55 Ma-3x and PI-1x simulations.





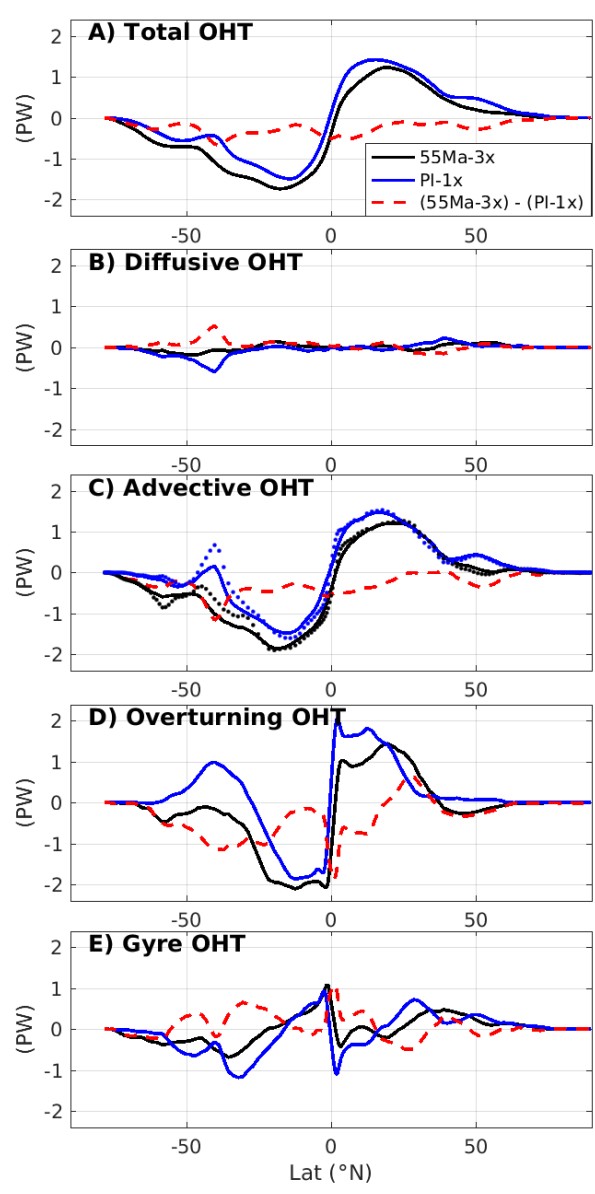

**Figure 7.** Meridional oceanic heat transport (in PW) as a function of latitude (positive contribution is northward) and its decomposition according to Eqs. 2 and 3. Results from the 55 Ma-3x and the PI-1x runs are shown in black and blue, respectively, and the difference between the two is in red. On panel C, the dotted lines indicate the sum of $OHT_{MOC}$ and $OHT_{gyre}$ estimated from monthly means, while the solid lines are computed at the model time step.



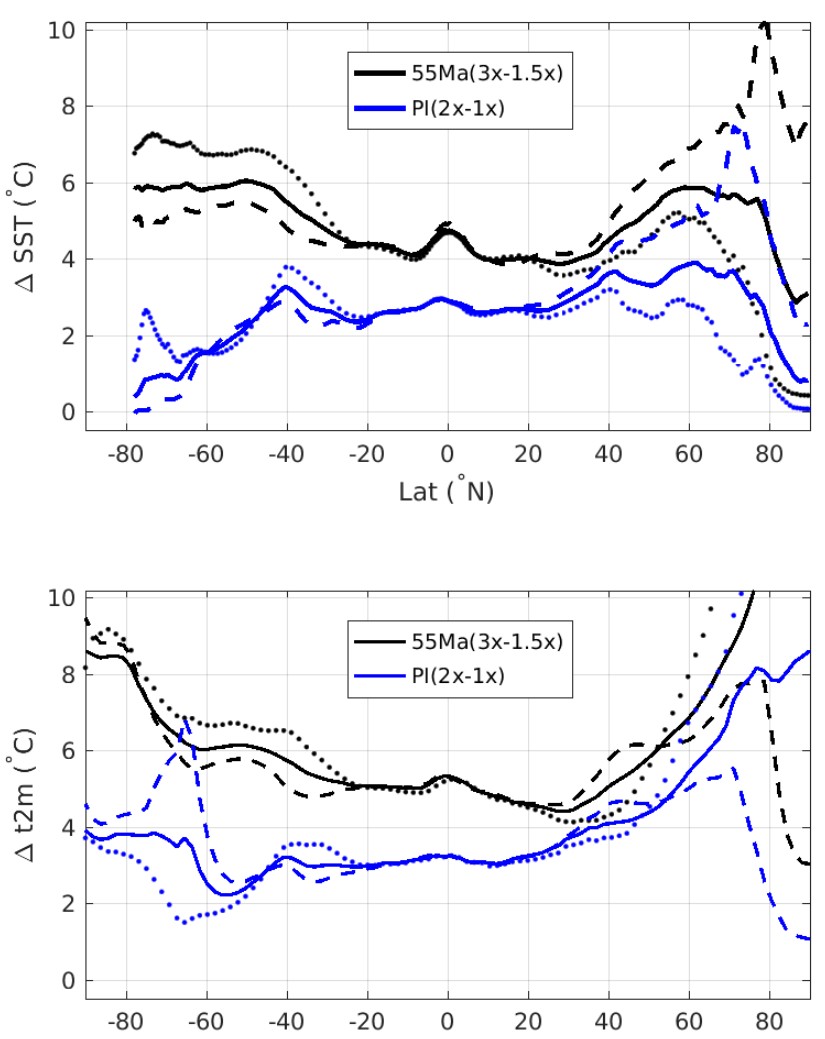

**Figure 8.** Difference of the zonally-averaged SST (top, in °C) and air temperature at 2m (bottom, in °C) as a function of latitude between the 55 Ma-3x and the 55 Ma-1.5x runs in black, and the PI-2x and PI-1x in blue. The solid line indicates the annual mean, the dashed line the mean over July-August-September, and the dotted line the mean over January-February-March.



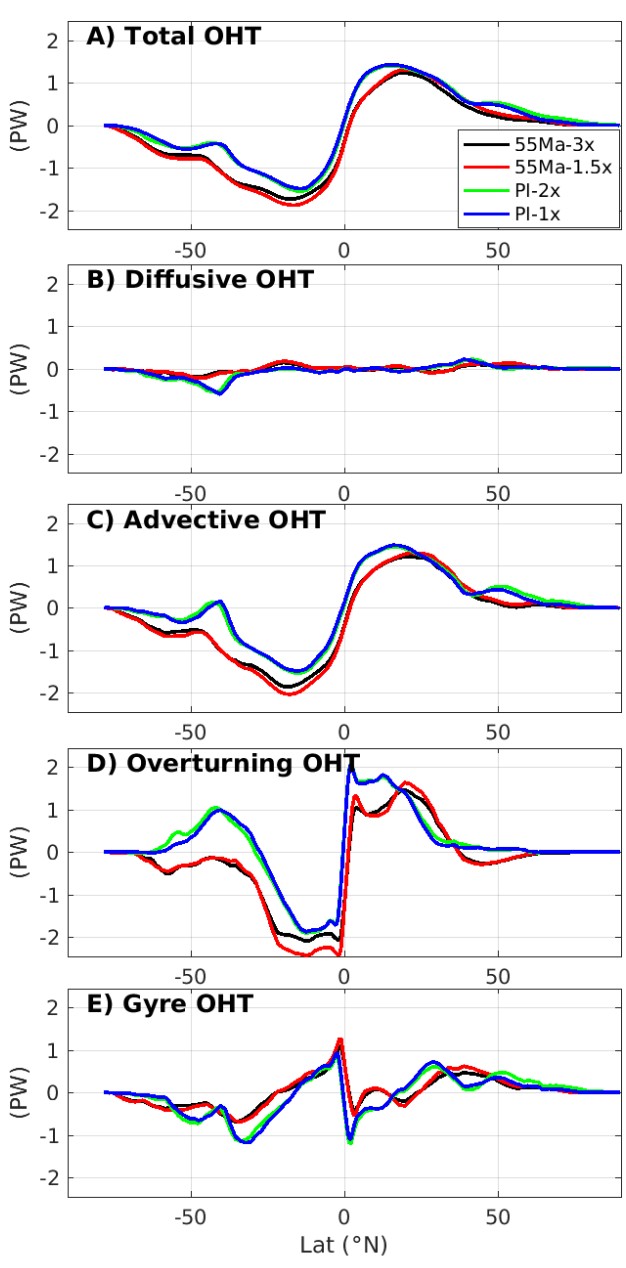

**Figure 9.** Meridional ocean heat transport (in PW) as a function of latitude (positive contribution is northward) and its decomposition according to Eqs. 2 and 3 in the 4 simulations.