# Peer review of "Early Eocene vigorous ocean overturning and its contribution to a warm Southern Ocean"

_Climate of the Past, 2019_

## Referee Comment (RC1) · Dan Lunt (Referee) · 20 Feb 2020

I reviewed this paper for another journal. The CP editor informed me that this paper is unchanged from that version, so I am attaching here my original review (as such the line numbers refer to the previous version).

Please also note the supplement to this comment:
https://www.clim-past-discuss.net/cp-2019-163/cp-2019-163-RC1-supplement.pdf

---

## Referee Comment (RC2) · Anonymous Referee #2 · 20 Mar 2020

I also reviewed the paper for another journal and the line numbers below refer to that version.

The study by Zhang et al. presents new model simulations of the early Eocene based on paleo-geographic reconstructions from the DeepMIP initiative, using the AOGCM IPSL-CM5A2. A comparison to simulations with modern conditions is used to investigate to role of ocean circulation changes and meridional heat transport for high latitude warmth in the Southern Hemisphere. The authors find a strong abyssal overturning circulation in the Southern Hemisphere (SOMOC) that leads to enhanced poleward heat transport that maintains warm Southern Hemisphere high latitudes. Additionally, in contrast to earlier studies no deep water is formed in the Northern Hemisphere in the early Eocene simulations. These are novel points worth to publish and the authors
provide an elaborate analysis of factors (e.g. tidal mixing scheme and CO2 changes) controlling the vigorous SOMOC in their early Eocene simulations. Nevertheless, the current manuscript could be improved by testing their findings in the context of North Pacific deep-water formation as this might be the most important modulator of the presented results. In that sense, the absence of North Pacific deep-water formation might be inherently linked to the basic question 'What does explain such a strong SOMOC?' (line 275).

Comments & Suggestions:

- To address this point, it might be helpful to include an additional experiment with a preindustrial CO2 concentration in the atmosphere. If this scenario is unlikely to give North Pacific deep-water formation, an artificial modification of the continental run-off distribution in the Pacific might help.

- Line 234: In the Weddell Sea surface density is much larger than in the North Pacific (+0.51 kg/m3). Is it trivial from this value that North Pacific deep-water formation is absent? Do you expect a critical value?

- It is recommended to include the CO2 sensitivity into the section '3.3 Factors contributing to the vigorous SOMOC'

- Please revise Table 2: the mean values and the relation to the uncertainty ranges is unclear, use units within the table for clarity, is the uncertainty range one or two sigma? Summary: Although the study is already at a reasonable level, the authors are encouraged to address the potential influence of North Pacific deep water on the presented findings. I would be happy to have a look at a revised manuscript version.

---

## Author Comment (AC1) · 14 Apr 2020

Dan Lunt (Referee)

I reviewed this paper for another journal. The CP editor informed me that this paper is unchanged from that version, so I am attaching here my original review (as such the line numbers refer to the previous version).

Please also note the supplement to this comment:https://www.clim-past-discuss.net/cp-2019-163/cp-2019-163-RC1-supplement.pdf

We are grateful to the reviewer for his insightful comments. We apologize about it but there has been some miscommunication between the editor, the reviewers and ourselves. Indeed, the submitted version of the paper to Climate of the Past was modified from the version the reviewer has previously reviewed, and most comments and suggestions were taken into account. Below we provide answers to the questions raised by the reviewer, i.e. we explained how and where these have been incorporated in the manuscript and clarified some points (we denoted the replies by blue color). The line numbers (in brackets) correspond to this revised version of the paper. These modified places are also marked by color in the annotated copy of manuscript.

In this paper, Zhang et al present results from IPSL-CM5A2 configured for the Eocene, and compare them with results of the modern. The focus is on ocean circulation, including regions of deep water formation, and the partitioning of heat transports into separate terms. Finally, sensitivity results to modified CO2 are presented. The paper is very well written and clear in general, and in my opinion it is very appropriate for Climate Dynamics, with minor-moderate revisions.

**General Comments**

Line 100-127. The model description and experimental design needs considerably more detail. Given that Sepulchre et al is just "in prep", we need many more details of the difference between IPSL-CM5A2 and IPSL-CM5A. For the experimental design, we need to know how soils, vegetation, sub-grid scale topography, and what frame of reference was used for the Herold reconstruction. The "bar" here is that the simulations should be approximately repeatable. The Lunt et al experimental design paper gives several options for many of the boundary conditions, so we need to know what choices have been made here.

This is a fair critic. First of all, since the initial submission, the paper by Sepulchre et al is now under review, and the preprint version is available from https://www.geosci-model-dev-discuss.net/gmd-2019-332/
Hence, we do not want to repeat too much of the material presented there. However, we have expanded largely the description of the model and of the different simulations, as our section 2.1 has been fully re-written. [line 90-166]

The model-data comparison needs some more work. In particular, the annual mean SSTs should be compared with the data in the main paper, and the seasonal in Supp Info, rather than the other way round.

We have moved the annual mean SST in the main text (Fig. 2A) and the summer SST in Supplementary information (Fig. S2A)

More explanation is needed of what the "uncertainty" in the proxies (Table 2) represents.

In response to this comment, we have added the uncertainty in the caption of Table 2, which is defined as follows: "The uncertainty range is defined as the $2\sigma$ deviations (2.2% and 97.8%) for $\delta^{18}O$, and as the range between 5% and 95% percentile SST estimates for $TEX^{86}$, Mg/Ca and clumped isotope data."

Also, it is claimed that the model-data agreement is "overall consistent", but this should be quantified more. For example, what RMS score would be obtained if it was assumed that the Eocene warmed uniformly, or with a cos(latitude) response. This can give some quantification to the question "how good is good". Also, the choice of 3x CO2 is rather arbitrary in the context of model-data comparison, so the GCM warming patterns could be scaled uniformly to best fit the data, and the RMS recalculated.

We are now presenting a more quantitative evaluation. First, we have justified better the fact that we mostly focus on the 55Ma-3x simulation, given that the agreement with the proxy-based SST reconstruction is better for this simulation than the other one (55Ma-1.5x simulation; see Table 2). Then, we have mostly followed the methodology proposed by Kennedy-Asser et al. (2019), that suggest 2 benchmarks to conduct such an evaluation. Benchmark 1 assumes a uniform mean temperature, and benchmark 2 is based on the assumption of the least squares linear fit through the proxy-based SST estimates with the cosine function of paleo-latitude from each-type-of-proxy sites. The results show the performance of the model simulation both RMSD metrics outperforms benchmark 1, but not benchmark 2 for some proxies. This means that the simulation outperforms the constant mean benchmark but not fully the latitudinal gradient benchmark, and thus it corresponds to the 'moderate good performance' following the wording used by Kennedy-Asser et al. (2019).

This is included in the text (lines 182-189), and Table 2, and in supplement materials (Lines 2-40, table S1 and figure 2B).

**Specific Comments**

Abstract - please mention the results of the CO2 sensitivity here. Otherwise "different levels of atmospheric CO2" on line 16 is hard to understand when reading for the first time.

The sentence 'Simulations with different atmospheric $CO_2$ levels show that the ocean circulation and heat transport are relatively insensitive to $CO_2$-doubling' has been added at the end of the abstract. [line 17-18]

Abstract – please add something about the model-data comparison that you have carried out.

We have added the sentence of "When compared with proxy-based reconstructions, the simulations reasonably capture both the reconstructed amplitude and pattern of early Eocene sea surface temperature." in the abstract. [line 4-5]

Line 19: Explain where (depth and latitude) that the 40Sv occurs.

The 40Sv MOC occurs at latitude of 60°S, as mentioned in the text. [line 8]

Line 32: Check whether 55 Ma is really the time period that DeepMIP focuses on; see e.g. Hollis et al (2019) and Lunt et al (2017).

We have modified this to 55Ma-50Ma, referring to the early-Eocene climate optimum (EECO) of DeepMIP framework, and added a sentence to clarify this. [Line 21, lines 24-26]

Line 40: I don't think it's correct that CO2 can't explain the decreased meridional gradient at the EECO. The high CO2 can lead to enhanced feedbacks at high latitudes.

The sentence has been changed to: 'In the early Eocene, high levels of $CO_2$ in the atmosphere are undoubtedly a critical contributor to the extremely warm climate, but they do not fully explain the extreme warmth at high-latitudes and the reduced equator-to-pole temperature gradient.' [line 29-32]

Figure SI1: For the time series of temperature in the simulations, the evolution seems inconsistent with the statement that the 1.5x run is branched off from the 3x run after 1500 years. Please align the simulations correctly so that the 1.5x run starts after 1500 years.

The figure has been modified, as indeed, "the 1.5x run is branched off from the 3x run after 1500 years".

As well as the time series of temperature (Figure SI1), it is important to show the time series of some metric of overturning, e.g. maximum overturning, or averaged mixed-layer depth, so we can assess to what extent the ocean circulation is in equilibrium, and the inter-annual variability of circulation.

We have included in Figure S1 the time series of MOC and temperature (both the SST and volume averaged temperature).

Figure 4: maybe use a log scale rather than using different scales for each panel, or use the same scale for each panel.

We had tried the log scale and the same scale for all panels by following this comment, but the figure becomes messy. We would like to keep it as it is to improve readability.

Line 223: Not clear what "intermittent" means here – it implies temporal variability.

We mean deepwater is formed during some winters. The sentence has been modified to improve clarity. [Line 255]

Line 234: I would have expected the vertical gradient in density, rather than surface density, to be the key control.

The density at depth does not change much, meaning that the change in vertical density gradient is largely determined by the surface density, which can be seen as a proxy. The text has been modified to explain this better. [line 263-265]

Line 237: For the salinity and temperature values, please also give the percentage change that this induces in density (e.g. 80% and 20%)

Change in salinity induces a density change of 0.57 kg/m$^3$ that is partly balanced by temperature-induced change of 0.06 kg/m$^3$. This is clarified in the text. [lines 266-268]

Line 243-254: I am not totally convinced by this mechanistic link to atmospheric circulation. Please either add some more quantitative analysis, or caveat this section with "possible" or "maybe".

This section has been fully revised to take into account this comment. [lines 274-285]

Line 294: This additional simulation should be introduced in the methods section.

This additional simulation (55Ma-3x-noM2) is now presented in the method section. [line 157-158]

Line 335-336: Please reference/add a Figure for "visible on sea surface salinity"

Sea surface salinity is partly decided by the freshwater budget (P-E+R) (Table 3) with the atmosphere. This budget is largely affected by the precipitation (Fig S3) when other processes are similar. So we showed the driver of sea surface salinity, precipitation, as an indicator, and added references on this. [lines 366]

Line 356-366: This section may link more clearly to ocean circulation if you explore the wind stress curl rather than the wind stress.

We had calculated the wind stress curl, which however is not as clear as the wind stress. So we would like to keep the wind stress for clarity. [lines 385-394]

Line 498-499: This non-linearity in climate sensitivity is interesting and could be explored in a bit more detail. See e.g. discussion of why this may be in Farnsworth et al, in press, GRL.

The larger temperature response of the 55Ma simulation to $CO_2$ doubling is consistent with the results of Farnsworth et al. (2019), that we now reference explicitly in the text. [lines 519-520]

Line 552: It is not clear what you mean by "robust".

We meant that the response is stable. Robust has been changed to stable. [line 565]

Line 560-570: Make clearer at the outset of this paragraph that it is a summary of the previous paradigm, not a summary of your results.

We have added "Numerous proxy based reconstructions have revealed that" in the first sentence to clarify the summary of previous studies, on which we started from. [lines 564-565]

**Technical Comments**

Line 37: "Ma ago" should be ""Ma".

Fixed. [line 28]

Line 44: "continental configuration" as well as "bathymetry"

Done [line 34]

Line 68: "latitude" should be "latitudes"

Done [line 56]

Line 82: remove "briefly"

Done [line 6-69])

Line 89: remove "IPSL in the following" and stick with the full name.

Done [line 75]

Equation 1: need to define E, W, x, z.

Done

Line 172: "clockwise" needs definition of which way we are "facing".

The meridional transport was integrated from the west to east across the basin, implying we are facing the west. [line 208-209]

Line 221: "deep convection" instead of "convection".

Done [line 253]

Line 242: net surface freshwater gain.

Fixed. [line 274]

Line 447: label the two terms in the equation, e.g. with a curly brackets, OHTmoc and OHTgyre.

We have labeled these two terms in the revised version. [line 471]

Line 530: HadCM3L.

Done [line 537]

Line 560: Warmest period in the Cenozoic.

Done [line 564-565]

Figure 2: Add labels for different lines.

Done

Figure 3: Add units for isopycnal contours in caption.

Done

---

## Author Comment (AC2) · 14 Apr 2020

I also reviewed the paper for another journal and the line numbers below refer to that version.

We are grateful to the reviewer for these insightful comments. As explained to the other reviewer the submitted version of the paper to Climate of the Past was modified from the version the reviewer has previously reviewed, and most comments and suggestions were already taken into account. Below we provide one-by-one replies to answer the questions raised by the reviewer (we denoted the replies by blue color) and to explain how and where these have been incorporated in the manuscript, and we done and/or to clarify some points. The line numbers (in brackets) correspond to the resubmitted version of the paper.

The study by Zhang et al. presents new model simulations of the early Eocene based on paleo-geographic reconstructions from the DeepMIP initiative, using the AOGCM IPSL-CM5A2. A comparison to simulations with modern conditions is used to investigate to role of ocean circulation changes and meridional heat transport for high latitude warmth in the Southern Hemisphere. The authors find a strong abyssal overturning circulation in the Southern Hemisphere (SOMOC) that leads to enhanced poleward heat transport that maintains warm Southern Hemisphere high latitudes. Additionally, in contrast to earlier studies no deep water is formed in the Northern Hemisphere in the early Eocene simulations. These are novel points worth to publish and the authors provide an elaborate analysis of factors (e.g. tidal mixing scheme and CO2 changes) controlling the vigorous SOMOC in their early Eocene simulations. Nevertheless, the current manuscript could be improved by testing their findings in the context of North Pacific deep-water formation as this might be the most important modulator of the presented results. In that sense, the absence of North Pacific deep-water formation might be inherently linked to the basic question 'What does explain such a strong SOMOC?' (line 275).

We have tried to elaborate a bit more in the revised version on why deep water formation occurs in the Southern Ocean but not in the Pacific, as explained in the following. [line 257-285]

Comments & Suggestions:

- To address this point, it might be helpful to include an additional experiment with a preindustrial CO2 concentration in the atmosphere. If this scenario is unlikely to give North Pacific deep-water formation, an artificial modification of the continental run-off distribution in the Pacific might help.
This is a good suggestion and indeed such an additional simulation (Eocene bathymetry and PI CO$_2$) would be useful. Yet, given the computational cost of running any of these simulations, it was not possible to run a new one for now (but we might in the future). However, the analysis of the Eocene 1.5-x simulation (which is not that different from what the reviewer is suggesting), and the comparison with the Eocene 1.5-x simulation suggests that the circulation is largely insensitive to the level of CO$_2$ during the Eocene (at least within this range), which makes us confident in the results presented in the manuscript.

- Line 234: In the Weddell Sea surface density is much larger than in the North Pacific (+0.51 kg/m3). Is it trivial from this value that North Pacific deep-water formation is absent? Do you expect a critical value?

What actually matters is the local change in stratification, of which the surface density is a good proxy in our case because the deep water density is much more similar in the two regions and between the different

simulations (we have clarified this point in the revised version, see Lines 261-266).  So, although it would be nice, we do not think there is such a thing as a threshold that allows (or not) for deep water formation in the Pacific. That said, such a statement should be checked more carefully, and could be a basis of a model intercomparison study based on the DeepMIP archive.

- It is recommended to include the CO2 sensitivity into the section '3.3 Factors contributing to the vigorous SOMOC'

We respectfully disagree with the reviewer on that point. We feel that the storyline of the paper would get messy if we were to include the $CO_2$ sensitivity results earlier on, and we would lose the coherency that we believe section 6 has as it is.

- Please revise Table 2: the mean values and the relation to the uncertainty ranges is unclear, use units within the table for clarity, is the uncertainty range one or two sigma?

Uncertainty range had been added in Table 2. It read as "The uncertainty range is defined as the $2\sigma$ deviations (2.2% and 97.8%) for $\delta^{18}O$, and as the range between 5% and 95% percentile SST estimates for $TEX^{86}$, Mg/Ca and clumped isotope data."

The units are now indicated in the table.

Summary: Although the study is already at a reasonable level, the authors are encouraged to address the potential influence of North Pacific deep water on the presented findings. I would be happy to have a look at a revised manuscript version.